# Neuro-KE: Scaling EEG Foundation Models via Label-Free Knowledge Integration

## Abstract

Foundation Models for Electroencephalography (EEG) have shown promise in learning generalized representations from large-scale datasets. However, current approaches primarily rely on raw signal reconstruction or scarce supervised labels, often neglecting the rich, domain-specific prior knowledge encapsulated in decades of signal processing research. In this work, we introduce Neuro-KE (Neuro-Knowledge Engine), a plug-and-play, label-free framework designed to seamlessly integrate comprehensive signal characteristics into the pre-training of EEG foundation models. Neuro-KE aggregates a 4-domain knowledge including Time Domain, Frequency Power, Frequency Structure, and Frequency Ratios features, distilling historical expertise into a unified knowledge base. We demonstrate the versatility and effectiveness of Neuro-KE across three mainstream technical paradigms: Masked Modeling, Contrastive Learning, and EEG–Large Language Models. Extensive experiments show that Neuro-KE significantly enhances model generalization and robustness, particularly in label-scarce downstream tasks, offering a rigorous pathway to embed domain-invariant signal dynamics into modern deep learning architectures.

## 1. Introduction

The advent of Foundation Models has revolutionized the landscape of artificial intelligence. These models demonstrate remarkable capabilities in generalization and transfer learning across Computer Vision (CV) and Natural Language Processing (NLP). Recently, this paradigm has begun to permeate the field of neuroscience. This shift has given rise to EEG Foundation Models designed to learn universal representations from massive archives of brain signals (Kuruppu et al., 2025; Ma et al., 2026; Wang et al., 2026; Chen et al., 2025). While the dominant approach remains Masked Modeling, which adapts architectures like BERT or MAE to reconstruct raw time-series (Wang et al., 2023; Zhang et al., 2023), emerging research is actively exploring alternative paradigms, such as contrastive learning (e.g., EEG-CLIP) (Ndir et al., 2025) and EEG-Large Language Models (e.g., NeuroLM) (Jiang et al., 2024a).

**Our Paradigm: Neuro-KE Guided Learning**

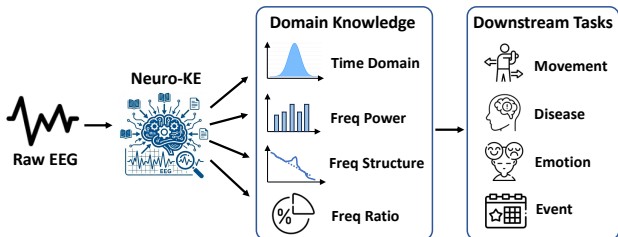

*Figure 1.* Scaling EEG Foundation Models via Neuro-KE.

Despite these rapid advancements, a fundamental challenge persists regarding the ambiguity of reconstruction targets in the neuro-signal domain. Unlike images or text, where models leverage the inherent "syntax", semantic structure of language tokens and human-interpretable structure (Chen et al., 2020; He et al., 2021). Raw EEG signals lack a known universal syntax and are often dominated by non-stationary noise (Sanei & Chambers, 2007). Purely data-driven reconstruction objectives often force models to expend significant capacity modeling stochastic noise or irrelevant artifacts(Zhang et al., 2023; Kuruppu et al., 2025) rather than capturing the underlying neural dynamics. This "semantic gap" limits pre-training efficiency and hinders alignment in multimodal settings where high-quality text-signal pairs are scarce.

To bridge this gap, we propose **Neuro-KE (Neuro-Knowledge Engine)**(Fig 1), a label-free framework that explicitly injects domain-specific prior knowledge into the foundation model learning process. By revisiting decades of signal processing research (Buzsáki, 2006; Sanei & Chambers, 2007; Al-Fahoum & Al-Fraihat, 2014), we construct a high-dimensional "Knowledge Manifold" comprising 62

[1]Anonymous Institution, Anonymous City, Anonymous Region, Anonymous Country. Correspondence to: Anonymous Author <anon.email@domain.com>.

Preliminary work. Under review by the International Conference on Machine Learning (ICML). Do not distribute.

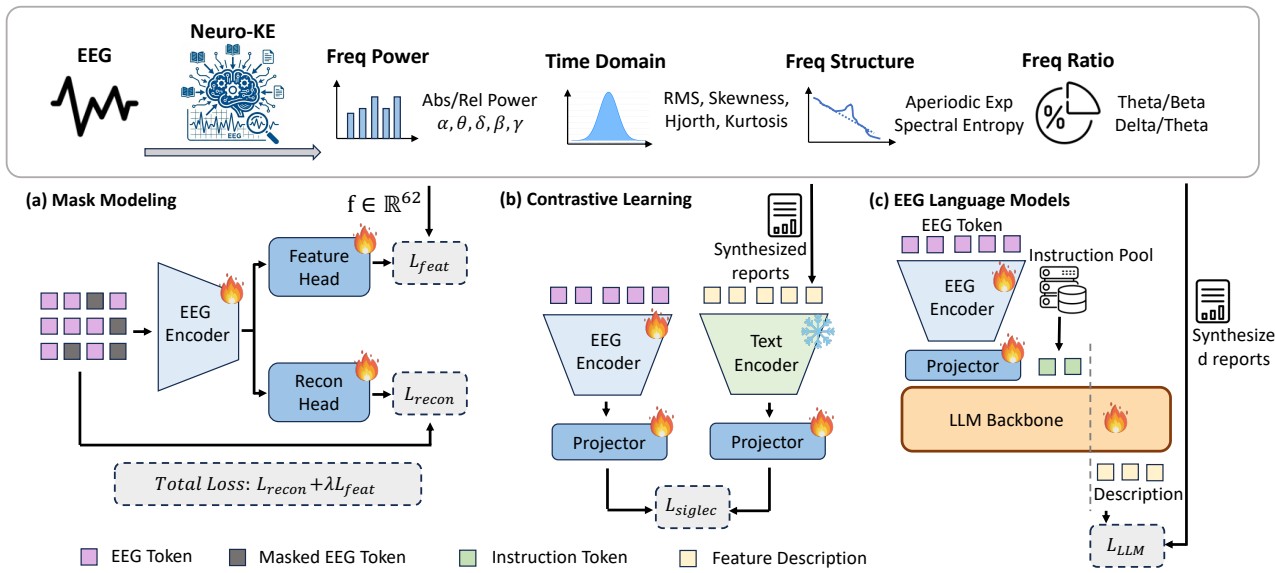

*Figure 2.* **Overview of the Neuro-KE Framework.** The core Knowledge Engine distills raw, unstructured EEG signals into a structured 4-domain knowledge manifold, capturing Time Domain statistics, Frequency Power, Frequency Structure, and Frequency Ratios. This manifold serves as a universal, plug-and-play interface tailored for three distinct foundation model paradigms via specific transformation strategies. (a) Masked Modeling: The feature vector serves as an auxiliary prediction target via a dedicated Feature Head. This is jointly optimized with the standard reconstruction loss, forcing the encoder to internalize physical signal dynamics within its latent space. (b) Contrastive Learning: Numerical features are serialized into synthesized clinical reports to generate semantic anchors. This enables label-free alignment between the EEG and Text encoders by treating synthesized descriptions as positive pairs for retrieval. (c) EEG-LLM: Features are formatted into structured Instruction QA Triplets. These are injected into the LLM backbone during instruction tuning, grounding the model's reasoning in objective signal characteristics.

distinct features across four complementary domains: time-domain statistics, frequency power distribution, spectral properties, and cross-frequency ratios. This approach functions as a universal, plug-and-play engine that powers three distinct paradigms: (a) acting as a semantic regularizer for Masked Modeling, (b) providing semantic anchors for Contrastive Learning, and (c) grounding reasoning in EEG–Large Language Models. This "Knowledge-Guided" approach captures critical domain-invariant dynamics often missed by purely data-driven methods.

Our contributions are three-fold:

(1) **Universal Knowledge Integration**: We introduce Neuro-KE, a framework that leverages a 62-feature manifold to ground models in physically meaningful properties without human annotation.

(2) **Unified Multi-Paradigm Validation**: We demonstrate that the same engine consistently enhances representation quality across Masked Modeling, Contrastive Learning, and EEG–Large Language Models.

(3) **Superior Data Efficiency**: Experiments confirm that Neuro-KE significantly boosts generalization; notably, in 1% low-resource regimes, our approach outperforms standard baselines by a remarkable margin.

## 2. Related Work

**From Manual Engineering to Deep Learning.** Early EEG analysis heavily depended on manual feature engineering, extracting spectral bands (Buzsáki, 2006), time-domain statistics (Al-Fahoum & Al-Fraihat, 2014), and non-linear complexity measures (Richman & Moorman, 2000) to interpret neural dynamics. Despite the rise of deep learning, these handcrafted features remain a dominant approach in clinical applications due to their exceptional robustness and stability across varying subjects and recording devices (Kemp et al., 2000; Dan et al., 2024). Recently, the field shifted toward end-to-end architectures—leveraging Convolutional Neural Networks (CNNs) (Lawhern et al., 2018; Schirrmeister et al., 2018) and Transformers (Song et al., 2022; Wan et al., 2023) to capture temporal dependencies directly from raw signals. While achieving state-of-the-art results on specific benchmarks (Chevallier et al., 2024), these data-driven models often require massive labeled datasets and struggle with distribution shifts in diverse real-world settings (Chau et al., 2024).

**EEG Foundation Models.** To mitigate label scarcity, the field has recently pivoted toward self-supervised Foundation Models, employing Masked Modeling (Wang et al., 2023; Zhang et al., 2023), generative pre-training (Yue

et al., 2025), and contrastive learning paradigms (Moham-madi Foumani et al., 2024). However, most existing approaches treat EEG as generic time-series, largely ignoring the rich biophysical constraints established in neuroscience.

**Knowledge Integration in Learning.** Bridging this gap, recent advances in CV and NLP have demonstrated the efficacy of structured priors (Chen et al., 2020) to improve sample efficiency and robustness. In the neuro-signal domain, initial attempts like DeeperBrain (Wang et al., 2026) have begun to incorporate "Neuro-Grounded" statistic prediction. However, a universal, label-free knowledge interface across paradigms remains unexplored.

Neuro-KE addresses this by formalizing a knowledge-guided framework that injects explicit time-frequency priors into modern architectures. A detailed review of these developments is provided in Appendix Section A.

## 3. Neuro-KE: The Neural Signal Knowledge Engine

### 3.1. The Neuro-KE Feature Space

The core of our proposed framework is the "Knowledge Feature Space", a high-dimensional representation $\mathbf{f} \in \mathbb{R}^{62}$ that encapsulates 62 distinct signal descriptors. Our feature selection is guided by three core principles to ensure robustness and scalability: **Physiological Interpretability**: prioritizing classic descriptors validated by extensive literature; **Computational Efficiency**: utilizing lightweight algorithms suitable for petabyte-scale extraction; and **Comprehensive Coverage**: spanning time, frequency, and non-linear domains to capture the full spectrum of neural dynamics. A complete definition of all 62 features is provided in Appendix Section C.

**Time Domain.** The Time Domain captures the statistical distribution and temporal complexity of the raw signal $x(t)$. This includes amplitude metrics like Root Mean Square ($RMS = \sqrt{\frac{1}{T}\sum x^2}$) and peak-to-peak amplitude (Al-Fahoum & Al-Fraihat, 2014), which serve as primary indicators of signal energy and artifact presence. We also incorporate distribution shape descriptors such as skewness and kurtosis (Sanei & Chambers, 2007), which are critical for detecting non-Gaussian transient events like interictal spikes. Furthermore, we employ Hjorth parameters (Activity, Mobility, Complexity) as computationally efficient proxies for spectral properties, enabling the model to sense frequency shifts directly in the time domain (Hu & Zhang, 2019).

**Frequency Power Domain.** The Frequency Power Domain quantifies the energy distribution across canoni-

cal frequency bands, computed via integration $P_{band} = \int_{f_1}^{f_2} P(f)df$. We compute both absolute and relative power for Delta (1-4 Hz), Theta (4-8 Hz), Alpha (8-13 Hz), Beta (13-30 Hz), and Gamma (30-45 Hz) bands, alongside total power (Buzsáki, 2006). Relative power features ($P_{rel} = P_{band}/P_{total}$) are particularly valuable as they normalize for inter-subject variability in skull thickness and conductivity, providing a more invariant representation of brain state.

**Frequency Structure Domain.** The Frequency Structure Domain characterizes the shape and topology of the Power Spectral Density (PSD), moving beyond simple band power integration. Key features include the aperiodic 1/f exponent $\chi$ (where $P(f) \propto 1/f^{\chi}$), which reflects the excitation-inhibition balance of the underlying neuronal population (Donoghue et al., 2020). We also calculate spectral entropy ($H = -\sum p(f)\log p(f)$) to measure signal complexity (Richman & Moorman, 2000), which often correlates with consciousness levels and cognitive load. Additionally, we identify oscillatory peaks such as the Individual Alpha Frequency (IAF) (Klimesch, 1999), a stable trait marker linked to information processing speed.

**Frequency Ratios Domain.** Finally, the Frequency Ratios Domain encodes cross-frequency interactions that serve as established biomarkers for complex cognitive states. Examples include the Theta/Beta ratio ($P_{\theta}/P_{\beta}$) for attention control and the Delta/Theta ratio for sleep depth analysis (Sanei & Chambers, 2007).

### 3.2. Knowledge Integration Strategies

To ensure Neuro-KE functions as a "plug-and-play" engine, we design three integration strategies tailored that leverage the same feature manifold as a universal interface (see Figure 2). Additional details on feature-guided data construction are provided in Appendix Section D.

**Latent Projection for Masked Modeling.** As illustrated in Figure 2(a), we map the feature vector $\mathbf{f} \in \mathbb{R}^{62}$ into the model's latent space using a lightweight Multi-Layer Perceptron (MLP) projection head $g_\phi$. This head transforms the encoder's latent representations $\mathbf{z}$ into a prediction $\hat{\mathbf{y}} = g_\phi(\mathbf{z})$, which is compared against the ground-truth feature vector $\mathbf{y}$. The auxiliary feature prediction loss is defined as the Mean Squared Error (MSE) between the predicted and actual feature vectors:

$$\mathcal{L}_{feat} = \frac{1}{N}\sum_{i=1}^{N} ||g_\phi(\mathbf{z}_i) - \mathbf{y}_i||^2 \qquad (1)$$

This constraint forces the encoder to recover physical properties alongside raw signal reconstruction (Wang et al., 2023).

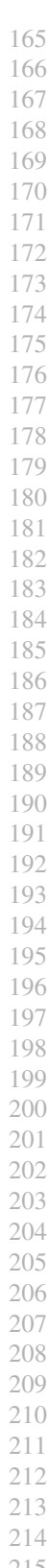

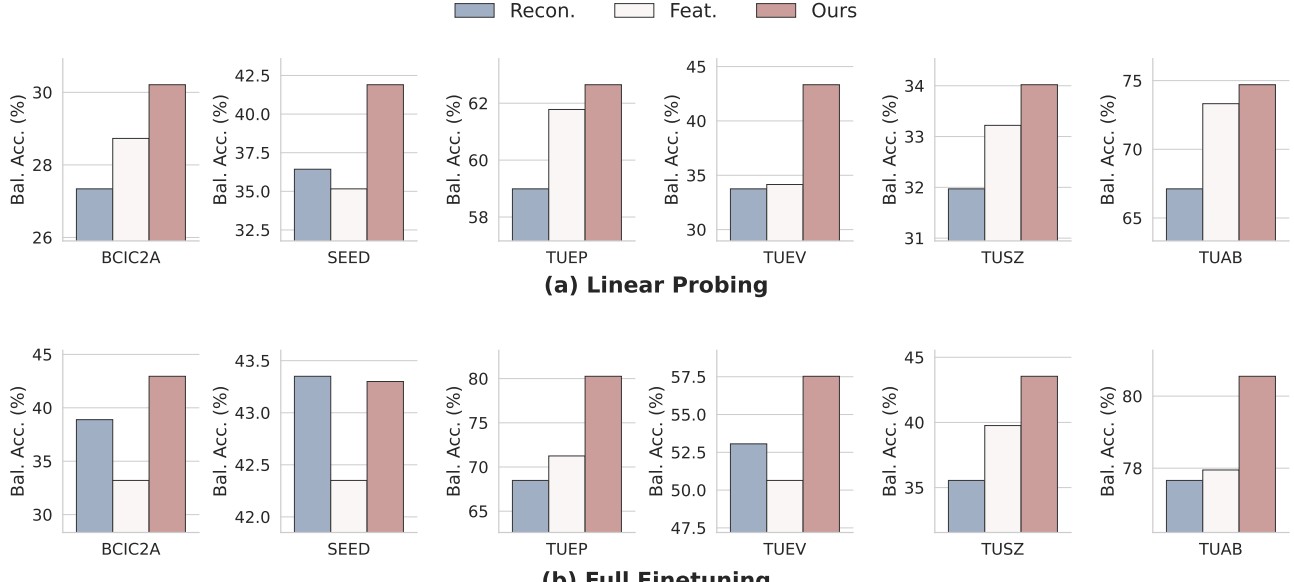

**(a) Linear Probing**

**(b) Full Finetuning**

*Figure 3.* **Cross-dataset performance of the Neuro-KE masked modeling approach. (a)** Linear probing results (Balanced Accuracy) across six EEG datasets. Our method, which augments masked signal reconstruction with auxiliary physical feature prediction, consistently outperforms both the reconstruction-only (*Recon.*) and feature-only (*Feat.*) baselines, indicating improved representation quality and transferability. **(b)** Full fine-tuning results on the same datasets, further confirming that explicitly enforcing physical signal constraints during pretraining yields robust and generalizable representations even when the entire model is updated.

**EEG-Text Pairs for Contrastive Learning.** As shown in Figure 2(b), we convert the numerical feature vectors into structured text templates using a grammar mapping $T = \text{Template}(\mathbf{f})$ (e.g., converting low alpha power to the sentence "The alpha power is attenuated (10-25th %ile)'). This generates synthetic text-EEG pairs. To align these modalities, we pass the EEG embeddings $E_{eeg}$ and Text embeddings $E_{text}$ through separate MLP adapters $h_e$ and $h_t$ to project them into a shared latent space. The embeddings are $L_2$ normalized: $\mathbf{v}_e = \frac{h_e(E_{eeg})}{||h_e(E_{eeg})||}$ and $\mathbf{v}_t = \frac{h_t(E_{text})}{||h_t(E_{text})||}$. We then optimize the InfoNCE loss to maximize the similarity between matched pairs:

$$\mathcal{L}_{NCE} = -\sum_i \log \frac{\exp(\mathbf{v}_{e,i}^\top \mathbf{v}_{t,i}/\tau)}{\sum_j \exp(\mathbf{v}_{e,i}^\top \mathbf{v}_{t,j}/\tau)} \qquad (2)$$

This process creates "semantic anchors" for robust alignment without requiring manual annotation (Ndir et al., 2025; Gijsen & Ritter, 2024).

**EEG-QA for EEG–Large Language Models.** As depicted in Figure 2(c), we format the feature sets into EEG-Question-Answer triplets $(x_{eeg}, x_{prompt}, A)$ where the answer $A$ is derived deterministically from the feature vector $\mathbf{f}$. The input prompt $x_{prompt}$ combines the EEG $x_{eeg}$, the query $x_{prompt}$, and the EEG context. The model is trained using the standard Next Token Prediction objective, minimizing the negative log-likelihood of the answer tokens

$A_t$:

$$\mathcal{L}_{LLM} = -\sum_{t=1}^{L} \log P(A_t|x_{prompt}, A_{<t}; \theta) \qquad (3)$$

This structured supervision grounds the Large Language Model's reasoning in objective signal characteristics (Jiang et al., 2024a; Lu et al., 2025).

## 4. Knowledge-Guided Masked Modeling

We apply Neuro-KE to the Masked Modeling paradigm to resolve the semantic ambiguity inherent in raw signal reconstruction (He et al., 2021). By treating the Neuro-KE Feature Space as a "semantic regularizer", we enforce physical consistency in the latent representation (Kostas et al., 2021). The backbone is a 12-layer CBraMod Transformer processes EEG as a sequence of non-overlapping patches(Wang et al., 2024b), alongside the standard masked reconstruction task (Wang et al., 2023). Detailed architectures and hyperparameters can be found in Appendix Section E.

**Experimental Setup.** We conducted a comprehensive evaluation on six diverse datasets, including TUAB (Abnormal), TUEV (Events), TUEP (Epilepsy), and TUSZ (Seizure) (Obeid & Picone, 2016), as well as SEED (Emotion) (Zheng & Lu, 2015) and BCIC2A (Motor Imagery) (Tangermann et al., 2012). Our evaluation protocol encompasses both Linear Probing (LP) and Full Fine-tuning (FT)

to rigorously assess representation quality. Implementation details regarding additional dataset-specific preprocessing described in Appendix Section B, feature-guided data construction and training configurations are provided in Appendix Sections D and E.

**Results and Performance Analysis.** As shown in Figure 3, Neuro-KE consistently outperforms the standard masked reconstruction baseline across all seven datasets. Under LP, joint optimization yields substantial gains, boosting balanced accuracy on TUAB from 67.12% to 74.70% and achieving a remarkable +9.6% improvement on the TUEV event detection benchmark (33.75% → 43.33%). Generalization extends beyond clinical tasks, with significant uplifts on SEED (Emotion) and BCIC2A (Motor Imagery). Under FT, the benefits amplify, particularly for pathology detection. Neuro-KE improves TUEP (Epilepsy) balanced accuracy from 68.48% to 80.26%. These results confirm that explicit biophysical priors are critical for recognizing complex epileptiform patterns, enhancing data efficiency and robustness across diverse regimes.

**Data Efficiency Analysis.** To validate the "knowledge scaffolding" hypothesis, we evaluated the model's performance in extreme low-data regimes by varying the training label ratio from from 0.5% to 100% on the TUAB dataset (see Figure 4). Under LP protocol, Neuro-KE demonstrates exceptional robustness, achieving ≈70% balanced accuracy with only 1% of training labels, significantly outperforming the reconstruction baseline (≈66%) and the feature-only baseline. This advantage persists in the FT protocol, where our method maintains a consistent lead across all data ratios. This result confirms that explicit knowledge injection provides a strong inductive bias, allowing the model to generalize effectively without needing massive supervised examples to rediscover basic signal properties.

**Feature Validation.** To verify that the encoder has internalized the domain knowledge rather than merely memorizing surface-level patterns, we evaluated its ability to reconstruct ground-truth features from frozen latent representations. As visualized in Figure 5, the model achieves high fidelity in feature prediction across all four domains. Notably, the density scatter plots reveal strong linear correlations ($r \geq 0.90$) for representative metrics such as Mean Zero-Crossing Rate ($R^2 = 0.76$) and Alpha Relative Power ($R^2 = 0.79$). Crucially, the model also successfully recovers complex, non-linear properties like the Aperiodic Exponent ($R^2 = 0.77$) and Theta/Beta Ratio ($R^2 = 0.77$), confirming that the pre-training process effectively encodes abstract physiological dynamics into the latent space. Detailed analyses can be found in Appendix Figures 9 to 11.

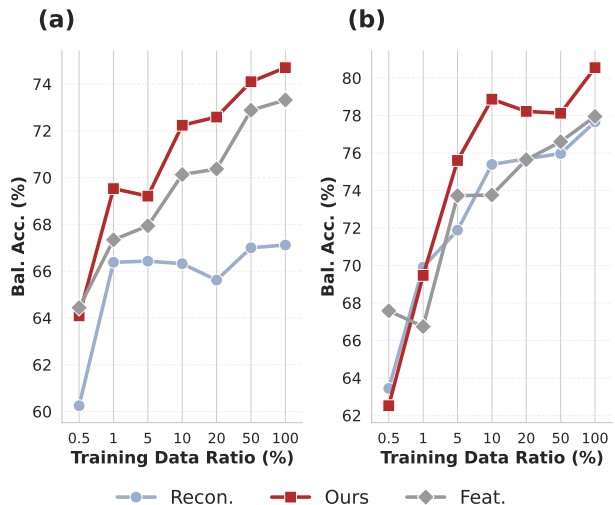

(a)      (b)

*Figure 4.* **Data Efficiency Analysis on the TUAB Dataset. (a)** Linear probing. **(b)** Full fine-tuning. The curves show Balanced Accuracy (%) as a function of training data availability (0.5% to 100%). Our method consistently achieves superior performance, particularly in low-data regimes, demonstrating that internalizing physical signal dynamics enhances label efficiency.

**t-SNE Visualization.** To provide qualitative insight into the learned representations, we visualize the latent embeddings of the TUAB dataset (Normal vs. Abnormal) using t-SNE . As illustrated in the Appendix Figure 12. As illustrated in the figure, the standard reconstruction baseline results in a highly entangled latent space, failing to distinguish between pathological and healthy states. In contrast, Neuro-KE exhibits distinct semantic separability, forming two clear, compact clusters corresponding to the clinical labels. The improved Silhouette Score confirms that by injecting domain knowledge, the encoder naturally disentangles biophysical semantics from background noise, effectively organizing the latent space even without supervised fine-tuning.

## 5. Knowledge-Driven Contrastive Learning

We leverage Neuro-KE to enable contrastive learning without requiring paired clinical reports, addressing the scarcity of high-quality text–signal supervision. Initializing from the robust encoder weights obtained in Paradigm 1, we employ a synthetic report generation pipeline on the TUAB dataset to construct a large-scale text–EEG corpus. Specifically, using the **Doubao-seed-1.8** API (Jiang et al., 2024a), we generate approximately 576,267 synthetic text–EEG pairs in total. Full generation procedures and template specifications are provided in Section D. To mitigate overfitting and prevent memorization of verbose reports, we adopt a positive sampling strategy that randomly pairs each EEG anchor with a single sentence sampled from its generated

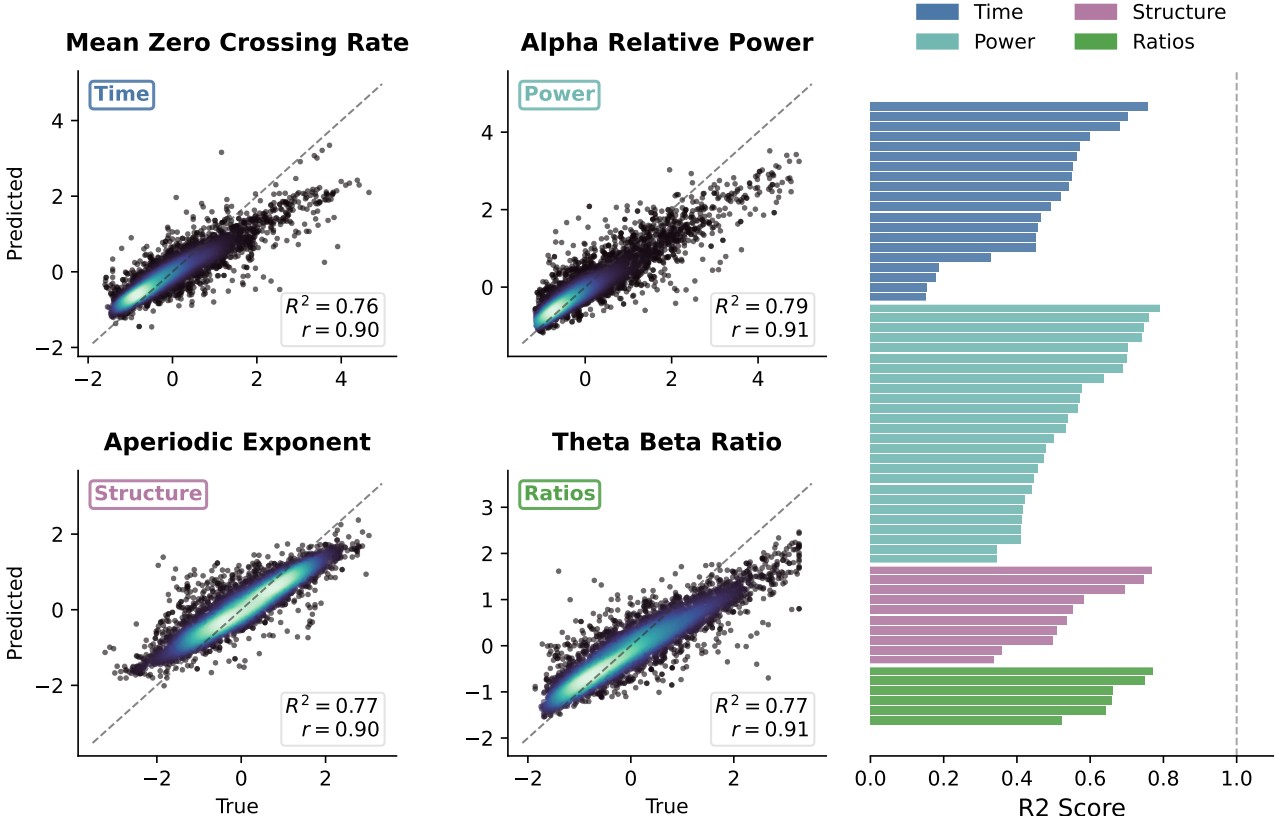

*Figure 5.* Performance analysis of EEG feature reconstruction. The left panel displays density scatter plots of predicted versus ground truth values for the top-performing feature within each of the four categories: Time (e.g., mean zero-crossing rate), Power (e.g., alpha relative power), Structure (e.g., aperiodic exponent), and Ratios (e.g., theta/beta ratio). The dashed diagonal line indicates perfect prediction ($y = x$), with $R^2$ and Pearson correlation ($r$) reported for each plot. The right panel summarizes the $R^2$ scores for all extracted EEG features, grouped by category and sorted by performance. The color coding corresponds to the feature categories: Time (blue), Power (teal), Structure (purple), and Ratios (green).

report pool (Radford et al., 2021; Ndir et al., 2025).

**EEG2Text Retrieval.** We evaluate Neuro-KE under the EEG2Text retrieval setting, where an EEG segment serves as a query to retrieve semantically aligned descriptions from the synthetic text corpus. As illustrated in Figure 6, we present two representative TUAB examples, including one Abnormal and one Normal recording. For each query, Neuro-KE retrieves highly relevant sentences that correctly reflect both the diagnostic label and salient physiological feature patterns. For instance, Abnormal EEG queries are matched with descriptions highlighting spectral irregularities and suppressed delta power, whereas Normal EEG queries retrieve statements indicating stable temporal dynamics. In contrast, mismatched texts from the opposite diagnostic class are assigned substantially lower or negative similarity scores. These qualitative retrieval results confirm that Neuro-KE learns a clinically meaningful shared embedding space, enabling label-free EEG-text alignment even without manually curated reports.

**Text-to-EEG Retrieval.** We further evaluate the reverse retrieval task, Text2EEG, where a textual description serves as a query to retrieve matching EEG segments from the signal database. As shown in Figure 7, we use two diagnostic prompts (*Normal* and *Abnormal*) as text anchors and retrieve the top-3 most relevant EEG excerpts for each query. The retrieved examples exhibit strong discriminability: abnormal prompts consistently return segments characterized by pathological dynamics, whereas normal prompts retrieve physiologically stable background activity. This clear separation demonstrates that Neuro-KE enables bidirectional EEG-text retrieval and establishes a robust shared embedding space grounded in structured neurophysiological semantics.

## 6. Knowledge-Enhanced EEG-Large Language Models

We introduce a three-stage curriculum to progressively equip the multimodal system with perceptual alignment, instruc-

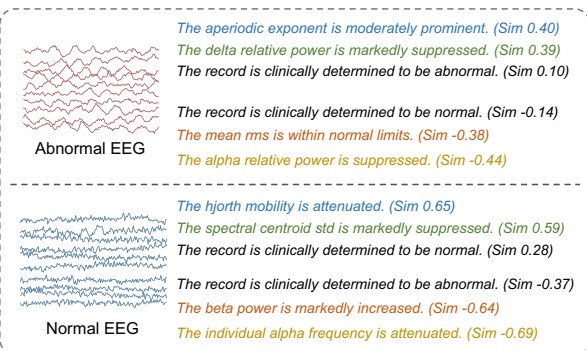

*Figure 6.* **EEG2Text retrieval examples enabled by Neuro-KE.** Given an EEG query segment, the model retrieves the most semantically aligned synthetic report sentences from the text corpus. We show one Abnormal (top) and one Normal (bottom) EEG example from TUAB. For each query, Neuro-KE ranks clinically consistent descriptions (e.g., abnormal aperiodic exponent or suppressed band power) with high similarity scores, while assigning low or negative similarity to mismatched diagnostic statements from the opposite class. These results demonstrate that Neuro-KE induces label-aware and physiologically grounded EEG-text alignment without paired clinical annotations.

tion following, and domain-specific reasoning. Our architecture integrates a Neuro-KE pre-trained EEG encoder, an MLP projector, and a Qwen3-0.6B LLM fine-tuned with LoRA (Hu et al., 2021). Across all stages, we construct approximately 364,867 EEG instruction triplets, with complete training hyperparameters and triplet generation templates provided in Sections D and E. In the first stage (Modality Alignment), we align the EEG feature space with the semantic space of the LLM using basic classification triplets $(x_{eeg}, x_{prompt}, y_{label})$. We initialize the EEG encoder with the robust weights from Paradigm 1. The projector is trained to minimize the Cross-Entropy loss on the label tokens:

$$\mathcal{L}_{Stage1} = -\sum_i \log P(y_{label}|Proj(E_{eeg}((x_{eeg})), x_{prompt})$$
$$(4)$$

In the second stage (Instruction Tuning), we unlock the reasoning capabilities by fine-tuning the entire pipeline on complex prompt templates. The objective is the standard Next Token Prediction loss over the answer sequence $A$:

$$\mathcal{L}_{SFT} = -\sum_{t=1}^{T} \log P(A_t|x_{eeg}, I, A_{<t}; \Theta)$$
$$(5)$$

Finally, the third stage (Knowledge Injection) injects explicit domain knowledge using two specialized triplet variants. The first trains on $(x_{eeg}, x_{prompt}, y_{report})$ to foster clinical reasoning, while the second uses $(x_{eeg}, x_{prompt}, y_{features})$ to ground the model in physical signal properties. We optimize the same Next Token Prediction objective on these knowledge-rich targets.

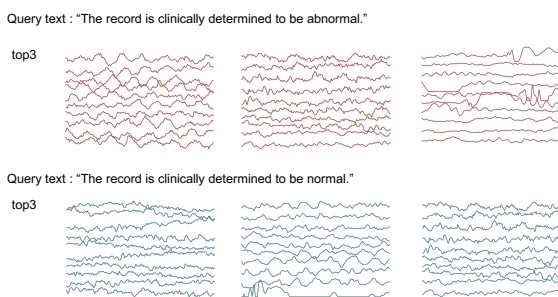

*Figure 7.* **Text2EEG retrieval examples enabled by Neuro-KE.** Given a diagnostic text query, Neuro-KE retrieves the most relevant EEG segments from the signal database. We show two prompts from TUAB (*Abnormal* and *Normal*), each retrieving the top-3 matching EEG excerpts. The retrieved signals exhibit clear physiological separation: abnormal queries consistently return pathological EEG patterns, whereas normal prompts retrieve stable background activity. These results highlight the bidirectional cross-modal alignment established by knowledge-guided contrastive learning.

**Classification Accuracy.** We first examine the impact of integrating Neuro-KE features into the EEG encoder for multimodal alignment. As shown in Table 12, the feature-enhanced CBraMod encoder substantially outperforms the original baseline during the initial Modality Alignment stage (Stage 1), achieving a classification accuracy of 79.0% compared to 70.4%. This large gap indicates that injecting structured signal priors during pre-training provides a superior initialization, enabling the projector to map EEG representations into the semantic space of the LLM more efficiently. While both models benefit from Stage 2 instruction tuning, the Neuro-KE variant maintains its advantage (81.3% vs. 80.0%). Moreover, explicit knowledge injection in Stage 3 further improves performance: the feature-guided variant achieves the highest overall accuracy of 83.6%, suggesting that grounding the model in distinct physical signal descriptors sharpens its decision boundary. The report-based variant also yields strong classification results, reaching 81.7%.

**Instruction Following and Clinical Reasoning.** Beyond binary label prediction, we evaluate whether the model can generate coherent feature descriptions and diagnostic narratives, as illustrated in Figure 8. We first test the Stage 1 aligned model's ability to respond to complex generative prompts requiring full reports or feature explanations. As shown in Table 13, the Stage 1 model largely fails to follow such instructions, producing valid responses in only 25% (reports) and 21% (features) of cases, with most outputs consisting of incoherent hallucinations or gibberish. In contrast, the Stage 3 models demonstrate robust instruction adherence and logical consistency. We measure coherence by verifying whether the generated explanation aligns with the predicted abnormality label. As reported in Table 14, the report-based model achieves an exceptional consistency

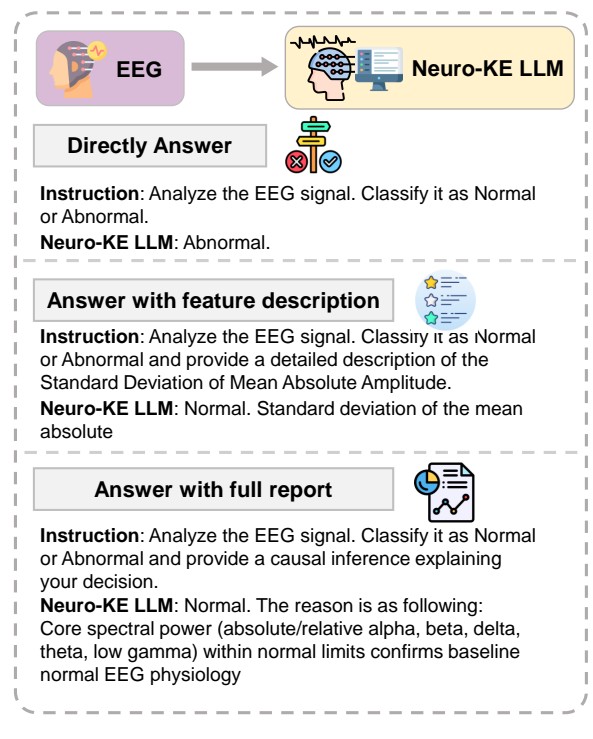

**Instruction**: Analyze the EEG signal. Classify it as Normal or Abnormal.
**Neuro-KE LLM**: Abnormal.

**Instruction**: Analyze the EEG signal. Classify it as Normal or Abnormal and provide a detailed description of the Standard Deviation of Mean Absolute Amplitude.
**Neuro-KE LLM**: Normal. Standard deviation of the mean absolute

**Instruction**: Analyze the EEG signal. Classify it as Normal or Abnormal and provide a causal inference explaining your decision.
**Neuro-KE LLM**: Normal. The reason is as following: Core spectral power (absolute/relative alpha, beta, delta, theta, low gamma) within normal limits confirms baseline normal EEG physiology

*Figure 8.* **Overview of the evaluation framework for Neuro-KE LLM.** The model processes EEG signals to perform three distinct diagnostic tasks: (1) Directly Answer, providing a binary classification of Normal vs. Abnormal; (2) Answer with Feature Description, which includes statistical details such as standard deviation; and (3) Answer with Full Report, generating a comprehensive diagnosis with causal inference and physiological reasoning.

rate of 99%, highly matching its classification decision with a clinically plausible narrative. The feature-based model also maintains high coherence, confirming that the multi-stage curriculum is necessary to bridge EEG perception with structured LLM reasoning grounded in neurophysiological knowledge.

## 7. Discussion & Conclusion

In this work, we introduced Neuro-KE, a pioneering "Knowledge Engine" that fundamentally rethinks the role of domain expertise in the era of Foundation Models. Our findings demonstrate that traditional feature engineering is not an obsolete practice, but a critical scaffold for modern deep learning (Li et al., 2025b). By distilling decades of signal processing research into a high-dimensional "Knowledge Manifold" of 62 descriptors, we provide a structured, physically meaningful vocabulary that bridges the semantic gap in EEG analysis (Buzsáki, 2006; Sanei & Chambers, 2007; Wang et al., 2023). Our validation across Masked Modeling, Contrastive Learning, and EEG-Large Language Models confirms that injecting these explicit priors signif-

icantly enhances data efficiency, generalization, and interpretability, particularly in challenging low-resource regimes where purely data-driven methods often falter (Kuruppu et al., 2025; Zhang et al., 2023).

Our findings advocate for a methodological shift toward a "Grey-Box" paradigm in neuroscience, where the interpretability of signal physics is fused with the scalability of deep learning to master complex biological signals (Wang et al., 2026; Serre & Pavlick, 2025; Zhang et al., 2025b). This paradigm has profound implications for global health equity, as the superior label efficiency of Neuro-KE lowers the barrier to entry for resource-constrained clinics lacking massive annotated archives (Obeid & Picone, 2016; Wu et al., 2025). Furthermore, by grounding model predictions in established biomarkers, such as linking seizure predictions to specific spike-wave variances, Neuro-KE fosters greater trust among clinicians and facilitates transparent decision support, addressing the critical "black box" hurdle in medical AI adoption (Ma et al., 2025; 2026).

Looking ahead, a key open question is how knowledge integration interacts with neural scaling laws, specifically, whether knowledge-guided models can achieve "emergence" at smaller parameter counts than their purely data-driven counterparts (Kaplan et al., 2020; Wang et al., 2024c). We plan to rigorously test this hypothesis by training Neuro-KE on multi-billion parameter backbones and petabyte-scale datasets, while simultaneously expanding the framework to encompass fMRI and MEG for a unified "Brain Foundation Model"(Zhou et al., 2025; Xiao et al., 2025; Dong et al., 2025). Finally, the computational efficiency of our interpretable features positions Neuro-KE as an ideal candidate for closed-loop Brain-Computer Interfaces, paving the way for adaptive neurofeedback and responsive neurostimulation on edge devices (Dan et al., 2024; Gao et al., 2025).

## Software and Data

The Neuro-KE framework, pre-trained model weights, and code for all downstream tasks will be made publicly available upon acceptance. The datasets used (TUEG, TUAB, etc.) are publicly accessible via the Temple University EEG Corpus and other repositories.

## Impact Statement

This work advances the capability of foundation models for EEG analysis, with potential applications in clinical diagnosis and BCI. We acknowledge the privacy implications of training on large-scale biosignals and emphasize the importance of data anonymization and ethical review in deploying such systems. There are no immediate negative societal consequences, but future deployment should consider fairness and bias across diverse populations.

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

# A. Extended Related Work

## A.1. Traditional EEG Analysis and Deep Learning

Early EEG research relied heavily on manual feature engineering to interpret neural dynamics. Researchers extracted domain-specific descriptors such as spectral power bands (Buzsáki, 2006; Al-Fahoum & Al-Fraihat, 2014), time-domain statistics (Sanei & Chambers, 2007; Hu & Zhang, 2019), and non-linear complexity measures like entropy (Richman & Moorman, 2000; Nunez & Srinivasan, 2006). Despite the rise of deep learning, these handcrafted features remain the dominant approach in practical clinical and BCI applications due to their exceptional robustness (Trigka et al., 2022; Singh & Krishnan, 2023). Unlike end-to-end models that can be brittle to distribution shifts, explicit features exhibit remarkable stability across varying subjects, recording devices, and electrode montages (Chaddad et al., 2023). This invariance makes them indispensable for real-world scenarios where data heterogeneity is the norm rather than the exception. These features served as robust inputs for classical classifiers in applications ranging from sleep staging (Kemp et al., 2000; Alvarez-Estevez & Rijsman, 2020) to emotion recognition (Zheng & Lu, 2015). With the advent of deep learning, the field shifted toward end-to-end architectures that learn directly from raw signals. Convolutional Neural Networks (CNNs) like EEGNet (Lawhern et al., 2018) and ShallowConvNet (Schirrmeister et al., 2018) became standard baselines for decoding oscillatory patterns. Subsequently, Transformer-based models such as EEG-Conformer (Song et al., 2022), EEGFormer (Wan et al., 2023), and ST-Transformer (Song et al., 2021) were introduced to capture long-range temporal dependencies. While these data-driven approaches achieved state-of-the-art performance on specific benchmarks (Chevallier et al., 2024; Dan et al., 2024), they often require massive labeled datasets, which are scarce in the medical domain. Moreover, due to their limited generalization capabilities under distribution shifts, the practical performance of these end-to-end models is often significantly compromised in diverse clinical settings (Crockett & Messeri, 2025; Chau et al., 2024).

## A.2. EEG Foundation Models

To address the label scarcity bottleneck, the community has rapidly adopted diverse self-supervised learning paradigms to train Foundation Models. Masked Modeling remains the most dominant approach, adapting BERT-style objectives to reconstruct masked signal patches (Wang et al., 2023; Zhang et al., 2023; Kostas et al., 2021; Wang et al., 2024a). Beyond simple masking, researchers have explored a rich variety of technical routes to capture temporal and spatial dependencies. Generative approaches leverage autoregressive next-token prediction to model the sequential nature of brain waves, as seen in BrainGPT (Yue et al., 2025) and NeuroLM (Jiang et al., 2024a). Vector Quantization (VQ) techniques have also gained traction, discretizing continuous signals into codebook indices to facilitate token-based learning (Gui et al., 2024; Barmpas et al., 2025a; Talukder et al., 2025). Parallel efforts explore contrastive learning strategies to align EEG representations with other modalities or augmentations (Mohammadi Foumani et al., 2024; Wei et al., 2025; Balestriero & LeCun, 2025). Recent works have scaled these paradigms to larger datasets and more complex architectures, including BrainBERT (Wang et al., 2023), Brant (Zhang et al., 2023), and BIOT (Yang et al., 2023). More recently, researchers have investigated integrating EEG with LLMs to enable semantic understanding and reasoning (Yue et al., 2025; Jiang et al., 2024a; Ma et al., 2026; Wang et al., 2026). Innovations such as NeuroLM (Jiang et al., 2024a), BrainGPT (Yue et al., 2025), and LaBraM (Jiang et al., 2024b) demonstrate the potential of cross-modal alignment (Ndir et al., 2025; Li et al., 2025a; Zhang et al., 2025a). Comprehensive surveys (Kuruppu et al., 2025; Li et al., 2025b; Zhou et al., 2025; Lai et al., 2025) highlight the rapid proliferation of these models, listing over dozens of variants proposed in the last two years (Barmpas et al., 2025b; Dong et al., 2025; Hong et al., 2025; Huang et al., 2025; Lu et al., 2025; Xiao et al., 2025). Fundamentally, EEG Foundation Models shift the paradigm from task-specific supervision to task-agnostic universality, enabling the learning of general-purpose representations from massive unlabeled corpora. This transition unlocks superior generalization capabilities, allowing models to adapt to unseen subjects and diverse hardware configurations with minimal fine-tuning (Kuruppu et al., 2025; Zhou et al., 2025).

## A.3. Knowledge Integration in Learning

Despite the success of data-driven Foundation Models, there is a growing recognition of the value of integrating domain knowledge into the learning loop. In Computer Vision and NLP, structured priors have been shown to improve sample efficiency and robustness (Chen et al., 2020; Grill et al., 2020; Radford et al., 2021; Alayrac et al., 2022). In the EEG domain, however, most Foundation Models continue to treat the signal as a generic time-series, ignoring the rich biophysical constraints established by traditional neuroscience (Varela et al., 2001; Canolty et al., 2006; Singh & Krishnan, 2023). A notable exception is DeeperBrain (Wang et al., 2026), which introduces a "Neurodynamics Statistics Prediction" objective.

This approach explicitly trains the model to predict macroscopic order parameters, including Relative Spectral Power (Buzsáki, 2006), Phase-Locking Values (PLV) (Varela et al., 2001), Cross-Frequency Coupling (CFC) (Canolty et al., 2006), and Sample Entropy (Richman & Moorman, 2000). Such "Neuro-Grounded" designs demonstrate that incorporating biophysical inductive biases can significantly enhance representation robustness. While DeeperBrain focuses on these specific statistics as reconstruction targets, few works have leveraged the full comprehensive manifold of signal processing features across diverse learning paradigms. Neuro-KE fills this gap by formalizing a "Knowledge Engine" that injects explicit time-frequency priors (Pradeepkumar et al., 2025; Sandino et al., 2025) into modern architectures. This approach bridges the divide between classical feature engineering and modern representation learning, offering a unified framework for knowledge-guided foundation models (Ma et al., 2025; Zhao et al., 2025; Yin & Shin, 2025).

## B. Dataset Details

### B.1. Dataset Summary

We utilize the TUEG corpus for self-supervised pretraining and seven diverse datasets for downstream evaluation. Table 1 summarizes the key statistics and task definitions.

*Table 1.* Summary of Pretraining and Downstream Datasets.

| DATASET | USAGE | TASK TYPE | CLASSES | INPUT DURATION |
|---------|-------|-----------|---------|----------------|
| TUEG | PRETRAINING | MASKED MODELING | - | 60S |
| TUAB | DOWNSTREAM | ABNORMAL DETECTION | 2 (NORMAL/ABNORMAL) | 60S |
| TUEV | DOWNSTREAM | EVENT CLASSIFICATION | 6 (SPSW, GPED, ETC.) | 1S |
| TUEP | DOWNSTREAM | EPILEPSY DETECTION | 2 (EPILEPSY/NO EPILEPSY) | 60S |
| TUSZ | DOWNSTREAM | SEIZURE DETECTION | 13 (SEIZURE TYPES) | VARIABLE |
| SEED | DOWNSTREAM | EMOTION RECOGNITION | 3 (POS/NEU/NEG) | 2S |
| BCIC2A | DOWNSTREAM | MOTOR IMAGERY | 4 (L/R/FOOT/TONGUE) | 1S |

### B.2. Dataset Descriptions

**TUEG (Temple University EEG Corpus)** (Obeid & Picone, 2016). The TUEG corpus serves as our large-scale unlabeled pretraining set. It contains over 30,000 clinical recordings from diverse patients. Since we employ a self-supervised masked modeling paradigm, we do not utilize any diagnostic labels from this dataset.

**TUAB (Abnormal EEG)** (Obeid & Picone, 2016). A subset of the TUH corpus labeled for clinical abnormality. The task is to classify a recording as either "Normal" or "Abnormal" based on pathological features.

**TUEV (EEG Events)** (Obeid & Picone, 2016). This dataset contains annotations for specific epileptiform events. Classes include Spike and Wave (SPSW), Generalized Periodic Epileptiform Discharges (GPED), and Periodic Lateralized Epileptiform Discharges (PLED).

**TUEP (Epilepsy)** (Obeid & Picone, 2016). Focused on patient-level diagnosis. The goal is to distinguish between patients with epilepsy and those without, regardless of whether a seizure is occurring in the specific segment.

**TUSZ (Seizure Detection)** (Obeid & Picone, 2016). The standard benchmark for automated seizure detection. It requires identifying specific seizure types (e.g., Focal, Generalized) against background activity.

**SEED (Emotion Recognition)** (Zheng & Lu, 2015). A dataset for affective computing. Subjects watched movie clips to elicit Positive, Neutral, or Negative emotional states.

**BCIC2A (Motor Imagery)** (Tangermann et al., 2012). A classic BCI dataset. Subjects performed motor imagery tasks: imagining movement of the Left Hand, Right Hand, Both Feet, or Tongue.

### B.3. Experimental Setup

For all downstream tasks, we strictly adhere to a Cross-Subject evaluation protocol to ensure that the model is tested on subjects it has never seen during training, evaluating its ability to learn universal, subject-invariant EEG representations. We employ a consistent subject-wise splitting strategy across all datasets, allocating 80% of subjects for training, 10% for

validation (hyperparameter tuning), and holding out the remaining 10% for final testing. For datasets with pre-defined splits (e.g., BCIC2A and SEED), we follow the standard benchmarks to ensure comparability.

### B.4. Preprocessing

We apply a standardized preprocessing pipeline to all datasets. First, a bandpass filter from 0.1 Hz to 75 Hz is used to remove DC drift and high-frequency noise, accompanied by a notch filter at 60 Hz (or 50 Hz depending on dataset region) to eliminate power line interference. Subsequently, we perform per-channel z-score normalization, computed independently for each input segment according to:

$$\tilde{x}_{c,t} = \frac{x_{c,t} - \mu_{c,\text{seg}}}{\sigma_{c,\text{seg}} + \epsilon} \tag{6}$$

where $\mu_{c,\text{seg}}$ and $\sigma_{c,\text{seg}}$ denote the mean and standard deviation of channel $c$ within the current segment. Finally, we standardize all recordings to the International 10-20 system with 19 channels (FP1, FP2, F3, F4, C3, C4, P3, P4, O1, O2, F7, F8, T3, T4, T5, T6, FZ, CZ, PZ), where missing channels are interpolated or zero-padded as necessary.

## C. Neuro-KE Feature Definitions

This appendix provides the precise mathematical formulations and taxonomy for the 62 features constituting the Neuro-KE Feature Space. Let $x(t)$ denote the discrete EEG time-series of length $T$, sampled at frequency $f_s$. Let $P(f)$ denote the Power Spectral Density (PSD) of $x(t)$, computed via Welch's method or FFT.

We selected a comprehensive set of 62 EEG features designed to capture complementary aspects of neural activity. These features are categorized into four distinct domains: Time Domain, Frequency Domain: Power, Frequency Domain: Structure, and Frequency Domain: Ratios. This multi-view approach ensures coverage of both oscillatory (rhythmic) and aperiodic (1/f-like) components of the EEG signal, as well as their temporal dynamics.

For each feature, we computed both the mean (`mean_` or `_power`) and the standard deviation (`_std`) across the analyzed window to capture both the central tendency and the temporal variability of the neural signal.

### C.1. Time Domain Features (19 Features)

These features characterize the statistical distribution and temporal complexity of the raw signal $x(t)$. Features in this domain describe the statistical distribution and temporal complexity of the raw time-series signal, providing insights into signal amplitude and dynamic behavior without explicit frequency transformation.

C.1.1. STATISTICAL MOMENTS

- **Amplitude Metrics:** `mean_abs_amplitude`, `mean_rms` (Root Mean Square), and `mean_peak_to_peak` quantify the overall magnitude of the signal.

- **Distribution Metrics:** `mean_skewness` and `mean_kurtosis` describe the shape of the signal distribution, helping to identify non-Gaussian artifacts or transient events.

- **Dynamics:** `mean_zero_crossing_rate` reflects the "roughness" or dominant frequency content in the time domain; `mean_channel_std` measures signal variability.

- **Mean Absolute Amplitude (MAA):**

$$\text{MAA} = \frac{1}{T} \sum_{t=1}^{T} |x(t)| \tag{7}$$

- **Root Mean Square (RMS):**

$$\text{RMS} = \sqrt{\frac{1}{T} \sum_{t=1}^{T} x(t)^2} \tag{8}$$

- **Peak-to-Peak Amplitude:**

$$\text{PTP} = \max(x(t)) - \min(x(t)) \tag{9}$$

- **Skewness (3rd Moment):**

$$\text{Skew} = \frac{\frac{1}{T}\sum_{t=1}^{T}(x(t)-\bar{x})^3}{(\frac{1}{T}\sum_{t=1}^{T}(x(t)-\bar{x})^2)^{3/2}} \tag{10}$$

- **Kurtosis (4th Moment):**

$$\text{Kurt} = \frac{\frac{1}{T}\sum_{t=1}^{T}(x(t)-\bar{x})^4}{(\frac{1}{T}\sum_{t=1}^{T}(x(t)-\bar{x})^2)^2} - 3 \tag{11}$$

- **Zero Crossing Rate (ZCR):**

$$\text{ZCR} = \frac{1}{T-1}\sum_{t=1}^{T-1}\mathbb{I}[(x(t)\cdot x(t+1)) < 0] \tag{12}$$

C.1.2. HJORTH PARAMETERS

Hjorth parameters provide computationally efficient time-domain proxies for spectral properties. Let $\sigma_0^2$ be the variance of $x(t)$, $\sigma_1^2$ be the variance of the first derivative $x'(t)$, and $\sigma_2^2$ be the variance of the second derivative $x''(t)$.

- Normalized descriptors of signal properties: `hjorth_activity` (signal variance), `hjorth_mobility` (mean frequency), and `hjorth_complexity` (change in frequency). These are computationally efficient proxies for spectral properties.

- **Activity:**

$$\text{Activity} = \sigma_0^2 \tag{13}$$

- **Mobility (Approx. Mean Frequency):**

$$\text{Mobility} = \sqrt{\frac{\sigma_1^2}{\sigma_0^2}} \tag{14}$$

- **Complexity (Approx. Bandwidth):**

$$\text{Complexity} = \frac{\text{Mobility}(x'(t))}{\text{Mobility}(x(t))} = \sqrt{\frac{\sigma_2^2/\sigma_1^2}{\sigma_1^2/\sigma_0^2}} \tag{15}$$

### C.2. Frequency Domain: Power (26 Features)

These features quantify the energy distribution across canonical frequency bands. These features represent the standard quantitative analysis, measuring the energy distribution across canonical frequency bands.

- **Abs (Absolute Power):** The total energy in specific frequency bands: Delta (1-4 Hz), Theta (4-8 Hz), Alpha (8-13 Hz), Beta (13-30 Hz), and Low Gamma (30-45 Hz). Note that High Gamma is excluded due to the 200 Hz downsampling preprocessing.

- **Rel (Relative Power):** The proportion of power in a specific band relative to the total power. This normalization helps mitigate inter-subject variability in skull thickness and conductivity.

- **Total:** `mean_total_power` represents the integrated energy across the entire frequency spectrum.

- **Band Power Integration:** For a frequency band defined by $[f_{\text{low}}, f_{\text{high}}]$:

$$P_{\text{band}} = \int_{f_{\text{low}}}^{f_{\text{high}}} P(f)df \tag{16}$$

  - **Delta:** $1 - 4$ Hz
  - **Theta:** $4 - 8$ Hz

- **Alpha:** $8-13$ Hz
- **Beta:** $13-30$ Hz
- **Gamma:** $30-45$ Hz

- **Total Power:**

$$P_{\text{total}} = \int_0^{f_{\text{nyquist}}} P(f) df \tag{17}$$

- **Relative Power:**

$$P_{\text{rel, band}} = \frac{P_{\text{band}}}{P_{\text{total}}} \tag{18}$$

### C.3. Frequency Domain: Structure (11 Features)

These features characterize the shape, topology, and complexity of the power spectrum. Beyond standard band power, these features characterize the shape and structure of the Power Spectral Density (PSD), capturing physiological information often missed by band-limited analysis.

- **Shape (Aperiodic & Distribution):**

    - `aperiodic_exponent`: The slope of the 1/f background activity. Flatter slopes have been linked to increased excitation/inhibition ratio.
    - `spectral_entropy`: Measures the complexity or disorder of the power spectrum.
    - `spectral_centroid`: The "center of mass" of the spectrum, indicating the dominant frequency range.

- **Peak (Oscillatory Peaks):**

    - `peak_frequency`: The frequency with the highest power.
    - `individual_alpha_frequency`: The specific peak frequency within the alpha band, which varies across individuals and correlates with cognitive processing speed.

- **Spectral Centroid:** The center of mass of the power spectrum:

$$f_c = \frac{\sum_f f \cdot P(f)}{\sum_f P(f)} \tag{19}$$

- **Spectral Entropy:** Measures the disorder or flatness of the spectrum (normalized Shannon entropy). Let $p(f)$ be the normalized PSD: $p(f) = \frac{P(f)}{\sum P(f)}$.

$$\text{SpEn} = -\frac{1}{\log(N_f)} \sum_f p(f) \log p(f) \tag{20}$$

- **Aperiodic 1/f Exponent** ($\chi$)**:** Obtained by fitting a power-law function to the background spectrum $L(f)$:

$$L(f) = \frac{b}{f^{\chi}} \tag{21}$$

$$\chi = -\text{slope of } \log(P(f)) \text{ vs. } \log(f) \tag{22}$$

- **Peak Frequency:** The frequency corresponding to the local maximum in the PSD.

    - **Individual Alpha Frequency (IAF):** $\text{argmax}_{f \in [8,13]} P(f)$

## C.4. Frequency Domain: Ratios (6 Features)

Cross-frequency interactions used as cognitive biomarkers. Cross-frequency interactions that serve as established biomarkers for specific cognitive and physiological states.

- **Band Ratio:**
    - `theta_beta_ratio`: A marker often associated with attention control and arousal regulation.
    - `delta_theta_ratio`: Used in sleep and slow-wave activity analysis.
    - `low_high_power_ratio`: A general index of slowing vs. activation.

- **Theta/Beta Ratio (TBR):**

$$\text{TBR} = \frac{P_\text{theta}}{P_\text{beta}} \tag{23}$$

- **Delta/Theta Ratio (DTR):**

$$\text{DTR} = \frac{P_\text{delta}}{P_\text{theta}} \tag{24}$$

- **Low/High Power Ratio:**

$$\text{LHR} = \frac{P_\text{delta} + P_\text{theta}}{P_\text{alpha} + P_\text{beta} + P_\text{gamma}} \tag{25}$$

## C.5. Feature List Table

*Table 2.* Complete list of Neuro-KE features.

| INDEX | DOMAIN | SUB-DOMAIN | FEATURE NAME | DESCRIPTION |
|---|---|---|---|---|
| 0 | TIME | STATS | MEAN_ABS_AMPLITUDE | MEAN ABSOLUTE AMPLITUDE |
| 1 | TIME | STATS | MEAN_ABS_AMPLITUDE_STD | VARIABILITY OF ABSOLUTE AMPLITUDE |
| 2 | TIME | STATS | MEAN_CHANNEL_STD | STANDARD DEVIATION OF AMPLITUDE |
| 3 | TIME | STATS | MEAN_CHANNEL_STD_STD | VARIABILITY OF CHANNEL STD |
| 4 | TIME | STATS | MEAN_KURTOSIS | KURTOSIS OF DISTRIBUTION |
| 5 | TIME | STATS | MEAN_KURTOSIS_STD | VARIABILITY OF KURTOSIS |
| 6 | TIME | STATS | MEAN_PEAK_TO_PEAK | PEAK-TO-PEAK AMPLITUDE |
| 7 | TIME | STATS | MEAN_PEAK_TO_PEAK_STD | VARIABILITY OF PTP |
| 8 | TIME | STATS | MEAN_RMS | ROOT MEAN SQUARE AMPLITUDE |
| 9 | TIME | STATS | MEAN_RMS_STD | VARIABILITY OF RMS |
| 10 | TIME | STATS | MEAN_SKEWNESS | SKEWNESS OF DISTRIBUTION |
| 11 | TIME | STATS | MEAN_SKEWNESS_STD | VARIABILITY OF SKEWNESS |
| 12 | TIME | STATS | MEAN_ZERO_CROSSING_RATE | ZERO CROSSING RATE |
| 13 | TIME | STATS | MEAN_ZERO_CROSSING_RATE_STD | VARIABILITY OF ZCR |
| 14 | TIME | HJORTH | HJORTH_ACTIVITY | SIGNAL VARIANCE |
| 15 | TIME | HJORTH | HJORTH_ACTIVITY_STD | VARIABILITY OF ACTIVITY |
| 16 | TIME | HJORTH | HJORTH_MOBILITY | MEAN FREQUENCY APPROX. |
| 17 | TIME | HJORTH | HJORTH_MOBILITY_STD | VARIABILITY OF MOBILITY |
| 18 | TIME | HJORTH | HJORTH_COMPLEXITY | BANDWIDTH APPROX. |
| 19 | TIME | HJORTH | HJORTH_COMPLEXITY_STD | VARIABILITY OF COMPLEXITY |
| 20-31 | POWER | ABS | [BAND]_POWER | ABSOLUTE POWER (D/T/A/B/G) |
| 32-43 | POWER | REL | [BAND]_RELATIVE_POWER | RELATIVE POWER (D/T/A/B/G) |
| 44 | POWER | TOTAL | MEAN_TOTAL_POWER | TOTAL SPECTRAL POWER |
| 45 | POWER | TOTAL | MEAN_TOTAL_POWER_STD | VARIABILITY OF TOTAL POWER |
| 46 | STRUCTURE | SHAPE | APERIODIC_EXPONENT | 1/F SLOPE |
| 47 | STRUCTURE | SHAPE | APERIODIC_EXPONENT_STD | VARIABILITY OF SLOPE |
| 48 | STRUCTURE | SHAPE | SPECTRAL_CENTROID | SPECTRAL CENTER OF MASS |
| 49 | STRUCTURE | SHAPE | SPECTRAL_CENTROID_STD | VARIABILITY OF CENTROID |
| 50 | STRUCTURE | SHAPE | SPECTRAL_ENTROPY | SPECTRAL COMPLEXITY |
| 51 | STRUCTURE | SHAPE | SPECTRAL_ENTROPY_STD | VARIABILITY OF ENTROPY |
| 52 | STRUCTURE | PEAK | INDIVIDUAL_ALPHA_FREQUENCY | ALPHA PEAK FREQUENCY |
| 53 | STRUCTURE | PEAK | INDIVIDUAL_ALPHA_FREQUENCY_STD | VARIABILITY OF IAF |
| 54 | STRUCTURE | PEAK | PEAK_FREQUENCY | GLOBAL PEAK FREQUENCY |
| 55 | STRUCTURE | PEAK | PEAK_FREQUENCY_STD | VARIABILITY OF PEAK FREQ |
| 56 | RATIOS | BAND | DELTA_THETA_RATIO | DELTA/THETA POWER RATIO |
| 57 | RATIOS | BAND | DELTA_THETA_RATIO_STD | VARIABILITY OF DTR |
| 58 | RATIOS | BAND | LOW_HIGH_POWER_RATIO | LOW/HIGH FREQ RATIO |
| 59 | RATIOS | BAND | LOW_HIGH_POWER_RATIO_STD | VARIABILITY OF LHR |
| 60 | RATIOS | BAND | THETA_BETA_RATIO | THETA/BETA POWER RATIO |
| 61 | RATIOS | BAND | THETA_BETA_RATIO_STD | VARIABILITY OF TBR |

# D. Feature-Guided Data Construction

## D.1. Knowledge-Guided Masked Modeling: Feature Preprocessing

Features for this paradigm are extracted from the TUEG pretraining corpus. To ensure stable training convergence and mitigate the impact of non-Gaussian distributions (e.g., extreme skewness and kurtosis commonly found in spectral power), we implement a hybrid adaptive preprocessing pipeline. This pipeline processes each feature dimension independently through three sequential steps. First, in the distribution quality check, we evaluate the normality of each feature using Skewness ($S$) and Kurtosis ($K$), classifying a feature as "Normal" if $|S| \leq 2.0$ and $K \leq 7.0$, and "Skewed" otherwise. Second, we apply adaptive transformation based on this classification: for Normal features, we retain the raw values; for Skewed features, we first apply a 6-Sigma Truncation to clamp values within $[\mu - 6\sigma, \mu + 6\sigma]$ to remove extreme outliers, and subsequently apply a Log1p transformation ($y = \log(x + 1)$) if the values are non-negative to correct right-skewed

heavy tails. Finally, regardless of the previous steps, all features undergo global standardization via Z-Score Normalization ($z = \frac{x-\mu}{\sigma}$) to unify the feature space into a standard scale ($\mu = 0, \sigma = 1$). The final pretraining dataset comprises approximately 8872 hours of recordings.

**D.2. Knowledge-Driven Contrastive Learning: Synthetic Report Generation**

To address the scarcity of paired EEG-text data, we employ a synthetic generation pipeline on the **TUAB** dataset. We use the **Doubao-seed-1.8** API to convert the 62-dimensional Neuro-KE feature vector into comprehensive clinical reports. To facilitate efficient contrastive learning, we pre-compute the embeddings for these text descriptions using the **Doubao-embedding** API. This pre-computation is feasible because the text encoder remains frozen during the alignment phase, allowing us to cache the embeddings offline. Crucially, we adopt a **one-to-many** strategy: each EEG segment is paired with a pool of individual descriptive sentences derived from its generated report. This diversity prevents overfitting to specific phrasing and enriches the semantic signal. The final dataset comprises approximately 576,267 synthetic text-EEG pairs.

**Prompt Template.** [

[You are a Clinical EEG Auditor. Your task is to generate a **High-Density, Telegraphic Clinical Impression**.

**[INPUT DATA]** Segment ID: segment id **[CLINICAL CONTEXT]** context text

Ground Truth Label: primary label

**[TECHNICAL SUMMARY]** technical text

**[QUANTITATIVE FEATURES (Raw Values Percentiles)]** feature text

**[STRICT GENERATION RULES]**

1. **Style Format**:
   – **Telegraphic Style**: Drop articles (the", a") and filler phrases.
   – **Structure**: EXACTLY 3–5 concise, causal bullet points.
2. **Causal Logic Requirement (CRITICAL)**:
   – **Format**: [Feature Y] (Raw=A, Pct=B) correlates with [Diagnosis Z]."
   – **Strict Ground Truth Adherence**: Use ONLY the provided quantitative features and ground truth. NEVER hallucinate a diagnosis if Ground Truth is Unknown.
3. **Contrastive Learning Optimization**: Use specific medical terminology (e.g., triphasic waves", PLEDs").
4. **Content Constraints**: **FORBIDDEN PHRASES**: We observe", The patient", This recording shows".

**[OUTPUT TEMPLATE]** Generate ONLY the Causal Inference.

STRICTLY 3–5 concise, causal bullet points.

Do NOT use section headers.]

]

**D.3. Knowledge-Enhanced EEG-Large Language Models: Instruction Tuning Triplets**

We construct instruction-tuning data in the format of $(x_{eeg}, x_{prompt}, A)$ from the **TUAB** training set, where $x_{eeg}$ is the raw EEG signal segment, $x_{prompt}$ is the instruction (e.g., "Analyze the alpha rhythm in this segment" or "Is there any evidence of seizure activity?"), and $A$ is the target answer derived deterministically from the Neuro-KE features or the dataset labels. The final instruction tuning dataset comprises approximately 364,867 triplets derived from the TUAB training set. This rich supervision signal enables the model to learn complex reasoning capabilities grounded in signal physics.

**Task Categories.** We created triplets for three main categories:

1. **Feature Description:** "Describe the frequency distribution." $\rightarrow$ "The signal shows dominant alpha activity in the posterior channels..."

2. **Classification:** "Classify the sleep stage." $\rightarrow$ "This segment corresponds to N2 sleep."

3. **Reasoning:** "Explain why this is abnormal." → "The presence of generalized spike-and-wave discharges indicates epileptiform activity."

**Prompt Template.**

1. **Stage 1:** "Analyze the EEG signal and classify it as Normal or Abnormal."

2. **Stage 2:** "Analyze the EEG signal and classify it as Normal or Abnormal."

3. **Stage 3-feature:** "Analyze the EEG signal. Classify it as Normal or Abnormal and provide a detailed description of the {feature name}."

4. **Stag 3-Report:** "Analyze the EEG signal. Classify it as Normal or Abnormal and provide a causal inference explaining your decision."

# E. Model Architecture & Hyperparameters

## E.1. Masked Modeling Architecture

### E.1.1. MODEL ARCHITECTURE

The backbone of Neuro-KE is a 12-layer Transformer based on the `cbramod` architecture, designed to process high-dimensional EEG signals efficiently. The detailed specifications are listed in Table 3.

We employ a factorized attention mechanism and specialized embeddings. The **Criss-Cross Attention** module factorizes attention into **Spatial Attention** (across $C = 19$ channels) and **Temporal Attention** (across $N = 60$ time patches), reducing complexity from quadratic to linear with respect to sequence length. The embedding layer consists of three components: a **Temporal** embedding using a 3-layer 1D Convolution stack, a **Spectral** embedding using FFT-based frequency encoding, and a **Positional** embedding implemented via depthwise separable convolution.

*Table 3.* Encoder Specifications.

| COMPONENT | SPECIFICATION | DESCRIPTION |
|---|---|---|
| BACKBONE | TRANSFORMER (CBRAMOD) | HIERARCHICAL TRANSFORMER WITH CRISS-CROSS ATTENTION |
| MODEL DIMENSION | $d_{model} = 200$ | HIDDEN DIMENSION OF THE ENCODER |
| FEED-FORWARD DIM | $d_{ff} = 800$ | DIMENSION OF THE FEED-FORWARD NETWORK |
| ATTENTION HEADS | $H = 8$ | NUMBER OF PARALLEL ATTENTION HEADS |
| LAYERS | $L = 12$ | NUMBER OF TRANSFORMER LAYERS |
| PATCH SIZE | $P = 200$ | 1 SECOND AT 200 HZ |
| SEQUENCE LENGTH | $N = 60$ | 60 PATCHES (60 SECONDS) |

### E.1.2. TRAINING STRATEGY

We adopt a patch-level Bernoulli masking strategy with a probability of $p = 0.5$. The training objective is defined as $\mathcal{L}_{total} = \mathcal{L}_{recon} + \lambda \cdot \mathcal{L}_{feat}$. The Reconstruction loss computes the Masked MSE on the raw signal, while the Feature Prediction loss calculates the MSE on 62-dimensional z-scored features. The pre-training hyperparameters are summarized in Table 4.

*Table 4.* Pre-training Hyperparameters.

| Hyperparameter | Value |
|---|---|
| Optimizer | AdamW |
| Learning Rate | $3 \times 10^{-4}$ |
| Weight Decay | 0.05 |
| Batch Size | 512 ($4 \times 128$) |
| Epochs | 30 |
| Scheduler | Cosine Annealing ($\eta_{min} = 10^{-6}$) |
| Warmup | None |
| Gradient Clipping | 1.0 |
| Loss Weight ($\lambda$) | 2.0 |

### E.2. Contrastive Learning Architecture

The contrastive learning framework aligns EEG representations with textual descriptions using a dual-encoder architecture.

#### E.2.1. Model Architecture

The EEG encoder adopts the `CBraMod` backbone, consisting of a patch embedding layer and a Transformer encoder. The patch embedding module processes raw EEG signals using a sequence of 2D convolutions (kernels $1 \times 49$, $1 \times 3$, $1 \times 3$) combined with spectral features extracted via FFT. The Transformer encoder comprises 12 layers with 8 attention heads, a model dimension ($d_{model}$) of 200, and a feedforward dimension of 800. The text encoder utilizes pre-computed Doubao vision embeddings with a dimension of 2048. Both modalities are projected into a shared 256-dimensional latent space for alignment. The detailed architecture of the adapters and dimension transformations is presented in Table 5.

*Table 5.* Contrastive Adapter Architecture and Dimension Alignment.

| Pipeline | Input Source | Source Dim | Adapter Structure | Output Dim |
|---|---|---|---|---|
| **EEG Stream** | CBraMod | 200 | MLP ($200 \rightarrow 200 \rightarrow 256$) | 256 |
| **Text Stream** | Doubao | 2048 | MLP ($2048 \rightarrow 200 \rightarrow 256$) | 256 |

#### E.2.2. Training Strategy

The model is trained using the Sigmoid Loss for Language Image Pre-training (SigLIP) on the TUAB dataset. Key hyperparameters are detailed in Table 6.

*Table 6.* Paradigm 2: Contrastive Learning Hyperparameters.

| Hyperparameter | Value |
|---|---|
| Optimizer | AdamW |
| Learning Rate | $1 \times 10^{-4}$ |
| Weight Decay | 0.05 |
| Batch Size | 256 |
| Epochs | 20 |
| Scheduler | Cosine Annealing |
| Loss Function | SigLIP |
| Temperature | Initial 10.0 (Learnable) |
| Projection Dim | 256 |
| Dropout | 0.1 |

### E.3. EEG Language Model Architecture

E.3.1. MODEL ARCHITECTURE

The EEG-LLM system integrates the pre-trained encoder with a Large Language Model via a lightweight projector, as shown in Table 7.

*Table 7.* EEG-LLM System Configuration.

| MODULE | COMPONENT | DETAILS |
|---|---|---|
| **EEG ENCODER** | FEATURE-ENHANCED CBRAMOD | INITIALIZED FROM PARADIGM 1, FINE-TUNED |
| **PROJECTOR** | MLP ADAPTER | 2-LAYER MLP, MAPS $d_{eeg} = 200 \rightarrow d_{llm}$ |
| **LLM BACKBONE** | QWEN3-0.6B | LOADED IN BFLOAT16 |
| **LoRA CONFIG** | LOW-RANK ADAPTATION | $r = 16$, $\alpha = 32$, DROPOUT=0.05 |
| **TARGET MODULES** | ATTENTION LINEAR LAYERS | ALL LINEAR LAYERS IN ATTENTION MECHANISM |

E.3.2. TRAINING STRATEGY

The training proceeds in three progressive stages. In **Stage 1 (Alignment)**, we train the projector and fine-tune the encoder on classification triplets while keeping the LLM frozen. **Stage 2 (Instruction Tuning)** involves training the projector, encoder, and LLM (using LoRA) on classification triplets. Finally, **Stage 3 (Knowledge Injection)** continues the training of all components on knowledge triplets, such as reports or feature descriptions. The hyperparameters for the instruction tuning phase are provided in Table 8.

*Table 8.* Instruction Tuning Hyperparameters.

| HYPERPARAMETER | VALUE |
|---|---|
| **OPTIMIZER** | ADAMW |
| **LEARNING RATE** | $1 \times 10^{-4}$ (ALL MODULES) |
| **BATCH SIZE** | 24 PER GPU |
| **WARMUP** | 7.5% OF TOTAL STEPS |
| **SCHEDULER** | COSINE ANNEALING |
| **STAGE 3 EPOCHS** | 10 (FEATURE), 6 (REPORT) |

## F. Evaluation Metrics

To rigorously assess model performance across diverse BCI tasks, we employ the following standardized evaluation metrics.

### F.1. Balanced Accuracy (B-Acc)

We prioritize Balanced Accuracy as our primary metric to address the inherent class imbalance prevalent in clinical EEG datasets (e.g., seizure vs. non-seizure segments). It is defined as the arithmetic mean of per-class recall:

$$\text{B-Acc} = \frac{1}{K} \sum_{k=1}^{K} \frac{TP_k}{TP_k + FN_k}, \tag{26}$$

where $K$ denotes the number of classes. This metric ensures that minority and majority classes contribute equally to the final score.

### F.2. AUROC (Area Under the ROC Curve)

For binary classification tasks such as Seizure Detection (TUSZ) and Abnormal EEG Detection (TUAB), we report the Area Under the Receiver Operating Characteristic Curve (AUROC), which measures the model's discriminative ability across all possible decision thresholds. For multi-class tasks, we adopt a macro-averaged AUROC over all classes.

**F.3. Weighted F1-Score**

We report the class-weighted F1-score to jointly account for precision and recall while respecting class frequency distributions. This is particularly important for imbalanced event classification tasks such as TUEV, where rare pathological patterns (e.g., GPEDs) must be detected reliably without being overwhelmed by majority classes. The F1-score is defined as:

$$\text{F1} = 2 \cdot \frac{\text{Precision} \cdot \text{Recall}}{\text{Precision} + \text{Recall}}, \tag{27}$$

and the final reported score is computed as a weighted average across classes, with weights proportional to the number of samples per class.

**F.4. Pearson Correlation Coefficient (Pearson $r$)**

To further evaluate the quality of the learned representations in auxiliary feature prediction tasks (Paradigm 1), we report the Pearson correlation coefficient between predicted features and their ground-truth values. Pearson $r$ measures the linear association between two variables and is defined as:

$$r = \frac{\sum(y - \bar{y})(\hat{y} - \bar{\hat{y}})}{\sqrt{\sum(y - \bar{y})^2}\sqrt{\sum(\hat{y} - \bar{\hat{y}})^2}}, \tag{28}$$

where $y$ and $\hat{y}$ denote the ground-truth and predicted feature values, respectively. A higher Pearson correlation indicates stronger alignment between the learned latent representations and the underlying physical signal properties.

**F.5. R-Squared ($R^2$)**

In addition to correlation-based evaluation, we use the coefficient of determination ($R^2$) to quantify how well the predicted features explain the variance of the ground-truth targets:

$$R^2 = 1 - \frac{\sum(y - \hat{y})^2}{\sum(y - \bar{y})^2}. \tag{29}$$

An $R^2$ value approaching 1 indicates that the encoder has successfully captured domain-relevant information necessary for accurate feature reconstruction.

# G. Additional Results

*Table 9.* Linear Probing Results across Six Downstream Tasks (100% Training Data). We compare three settings: **Recon.** (Masked Reconstruction Baseline), **Feat.** (Feature Prediction Only), and **Ours** (Neuro-KE Joint Optimization). All experiments use 200-dimensional representations.

| DATASET | MODEL | BACC | AUROC | WEIGHTED F1 |
|---|---|---|---|---|
| **BCIC2A** | RECON. | 27.34% | 0.5251 | 0.2663 |
| (MOTOR IMAGERY) | FEAT. | 28.73% | 0.5552 | 0.2454 |
| | OURS | **30.21%** | **0.5645** | **0.2899** |
| **SEED** | RECON. | 36.43% | 0.5569 | 0.3372 |
| (EMOTION) | FEAT. | 35.16% | 0.5393 | 0.3173 |
| | OURS | **41.90%** | **0.6033** | **0.4222** |
| **TUEP** | RECON. | 58.99% | 0.6940 | 0.5996 |
| (EPILEPSY) | FEAT. | 61.78% | 0.6313 | 0.6359 |
| | OURS | **62.65%** | **0.7019** | **0.6431** |
| **TUEV** | RECON. | 33.75% | 0.8049 | 0.5207 |
| (EVENTS) | FEAT. | 34.15% | 0.7982 | 0.5182 |
| | OURS | **43.33%** | **0.8476** | **0.6108** |
| **TUSZ** | RECON. | 31.97% | 0.8480 | 0.6912 |
| (SEIZURE) | FEAT. | 33.22% | 0.8449 | 0.7088 |
| | OURS | **34.02%** | **0.8604** | **0.7195** |
| **TUAB** | RECON. | 67.12% | 0.7359 | 0.6724 |
| (ABNORMAL) | FEAT. | 73.32% | 0.8097 | 0.7344 |
| | OURS | **74.70%** | **0.8284** | **0.7484** |

*Table 10.* Full Fine-tuning Results across Six Downstream Tasks (100% Training Data). We compare three settings: **Recon.**, **Feat.**, and **Ours**. Note: TUEV and TUSZ do not report AUROC.

| DATASET | MODEL | BACC | AUROC | WEIGHTED F1 |
|---|---|---|---|---|
| **BCIC2A** | RECON. | 38.89% | 0.6502 | 0.3527 |
| (MOTOR IMAGERY) | FEAT. | 33.20% | 0.6021 | 0.3121 |
| | OURS | **42.95%** | **0.6793** | **0.4142** |
| **SEED** | RECON. | **43.35%** | **0.6051** | **0.4148** |
| (EMOTION) | FEAT. | 42.35% | 0.6003 | 0.4134 |
| | OURS | 43.15% | 0.6011 | 0.4108 |
| **TUEP** | RECON. | 68.48% | 0.7473 | 0.6857 |
| (EPILEPSY) | FEAT. | 71.25% | 0.8117 | 0.7266 |
| | OURS | **80.26%** | **0.8700** | **0.7987** |
| **TUEV** | RECON. | 53.06% | - | 0.6580 |
| (EVENTS) | FEAT. | 50.64% | - | 0.6549 |
| | OURS | **57.53%** | - | **0.7291** |
| **TUSZ** | RECON. | 35.55% | - | **72.62%** |
| (SEIZURE) | FEAT. | 39.76% | - | 65.69% |
| | OURS | **43.54%** | - | 67.16% |
| **TUAB** | RECON. | 77.66% | 0.8441 | 0.7654 |
| (ABNORMAL) | FEAT. | 77.95% | 0.8530 | 0.7612 |
| | OURS | **80.55%** | **0.8878** | **0.8047** |

*Table 11.* Comparison of prediction accuracy of Neuro-KE LLM between the Original cbramod and the Feature-Enhanced cbramod (Ours) across Stage 1 and Stage 2.

| Model Variation | Stage 1 | Stage 2 |
|---|---|---|
| Original model | 0.704 | 0.800 |
| Feature-Enhanced model (Ours) | **0.790** | **0.813** |

*Table 12.* Performance results of the Feature-Enhanced cbramod LLM in Stage 3.

| Model Variation | Stage 3 (Report) | Stage 3 (Feature) |
|---|---|---|
| Feature-Enhanced model (Ours) | 0.817 | **0.836** |

*Table 13.* Analysis of Response Validity in Stage 1. This metric measures the proportion of valid (non-hallucinated) responses when the model is prompted to generate reports or features alongside labels.

| Model Variation | Stage 1 (Report) | Stage 1 (Feature) |
|---|---|---|
| Feature-Enhanced model (Ours) | 25% | 21% |

*Table 14.* Logical Consistency Analysis in Stage 3. This measures the percentage of instances where the predicted label is consistent with the content of the generated report or feature.

| Model Variation | Stage 3 (Report) | Stage 3 (Feature) |
|---|---|---|
| Feature-Enhanced model (Ours) | 99% | 85% |

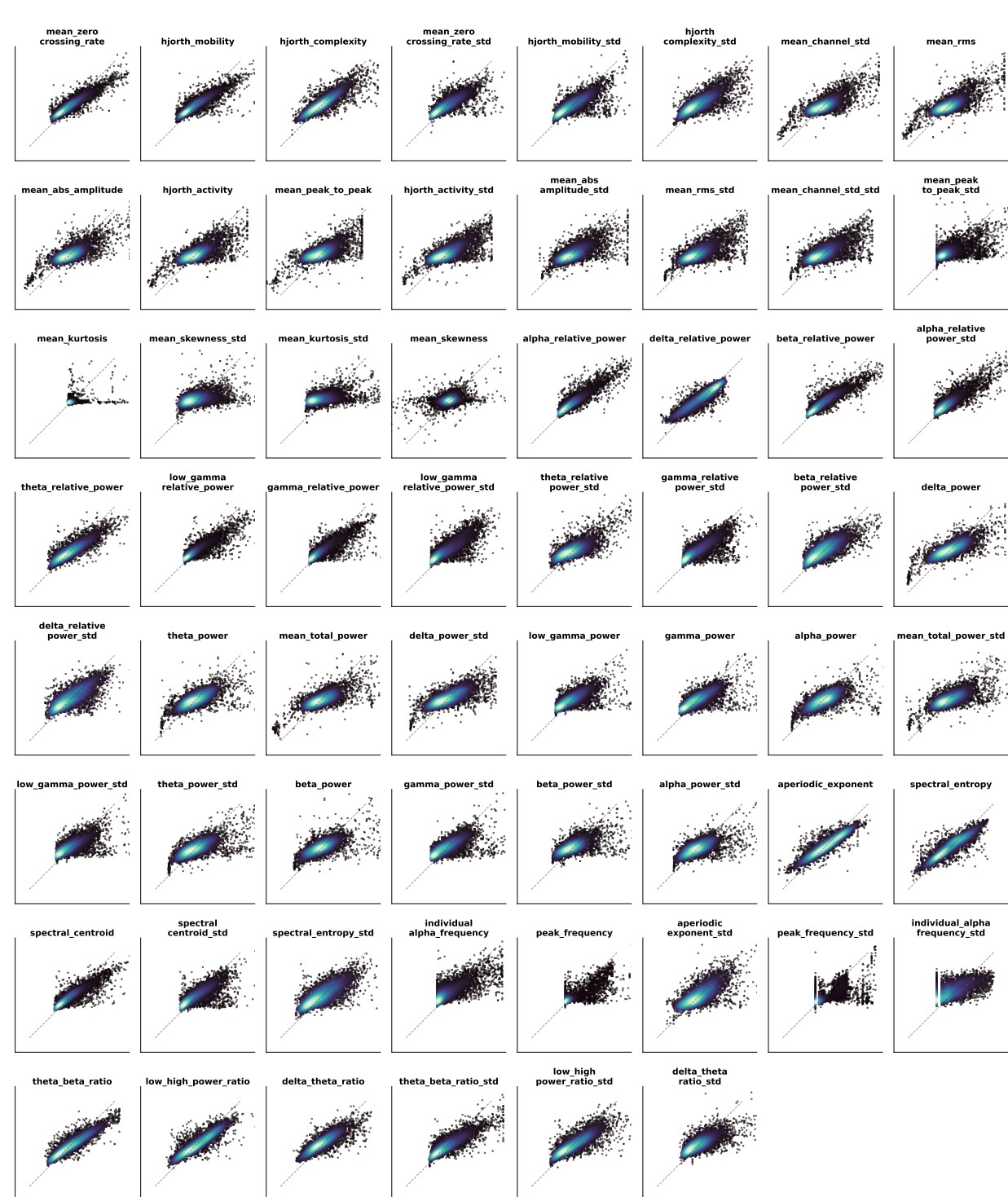

*Figure 9.* **Neuro-KE feature prediction accuracy.** Scatter plots comparing the predicted versus ground-truth values across all 62 Neuro-KE descriptors. Each subplot corresponds to one feature dimension, demonstrating strong correlation and accurate recovery of physiologically meaningful statistics. This validates that the encoder successfully internalizes structured signal priors during masked modeling pre-training.

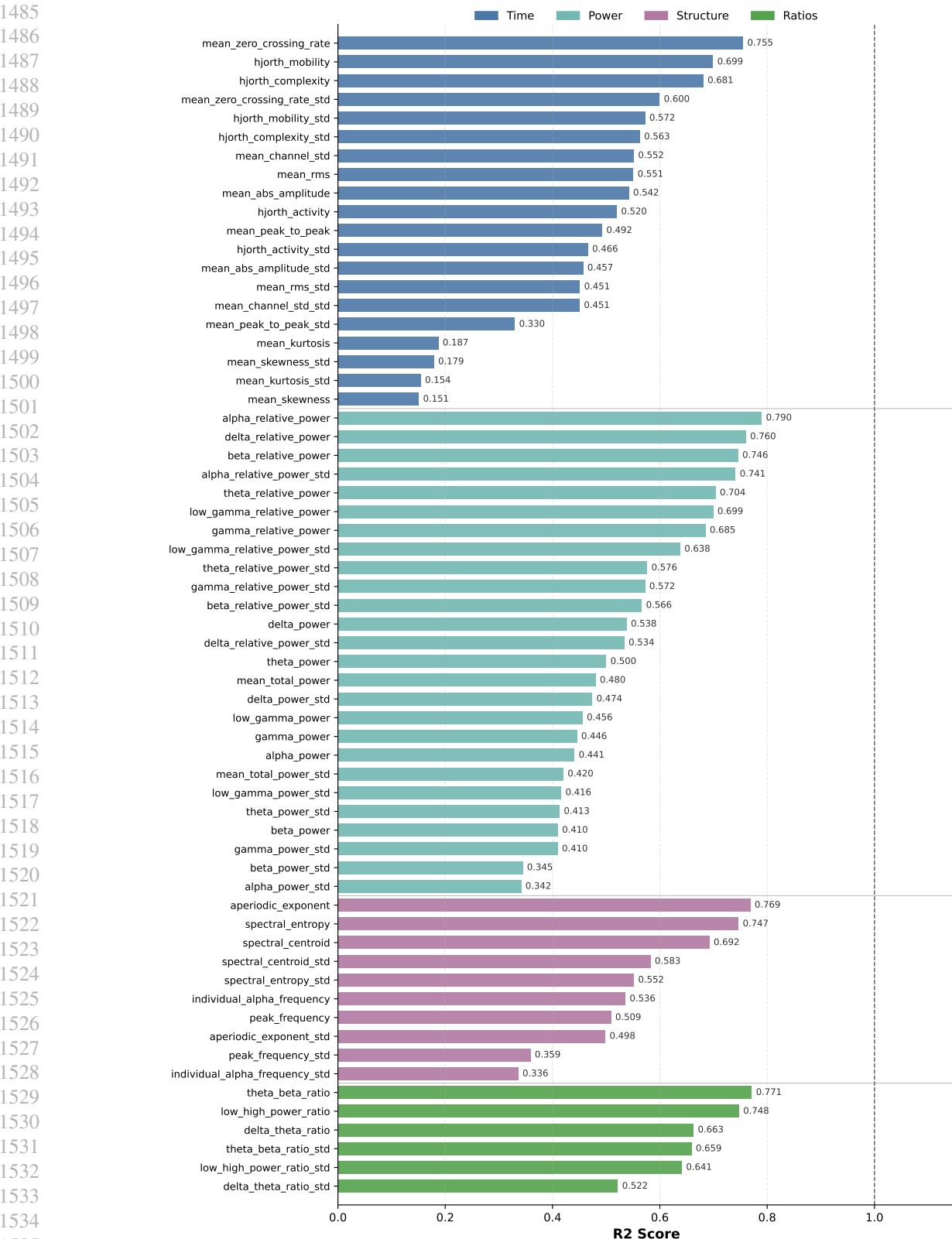

*Figure 10.* **Per-feature $R^2$ prediction performance across the Neuro-KE manifold.** We report the coefficient of determination ($R^2$) for each of the 62 signal descriptors grouped by domain: Time (blue), Frequency Power (green), Frequency Structure (purple), and Frequency Ratios (orange). Neuro-KE achieves consistently high predictability on physiologically meaningful features, particularly for spectral power and aperiodic structure descriptors, confirming that the learned latent space captures structured time–frequency priors beyond raw waveform reconstruction.

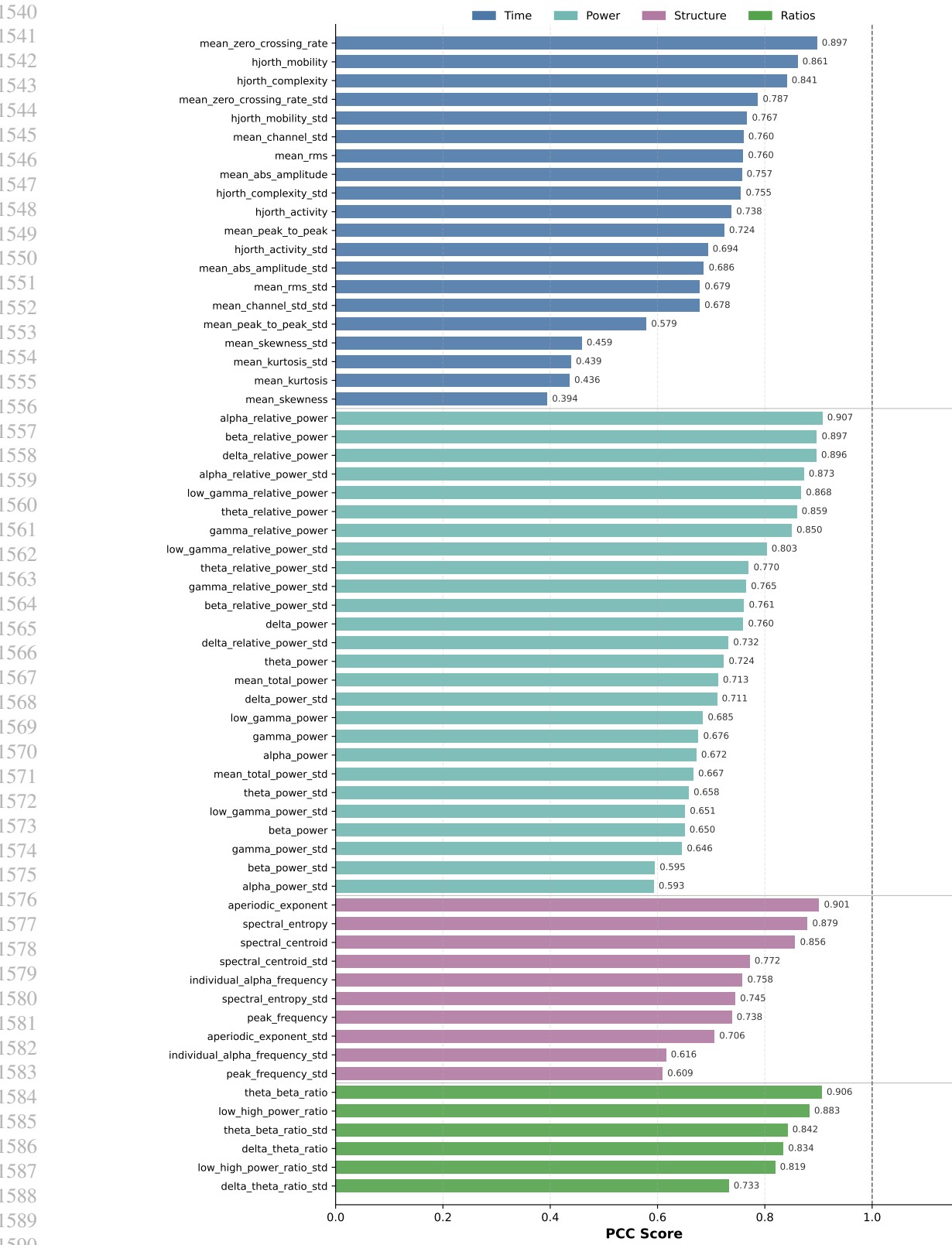

*Figure 11.* **Per-feature Pearson correlation (PCC) across the Neuro-KE manifold.** We report the Pearson Correlation Coefficient (PCC) between predicted and ground-truth values for all 62 Neuro-KE descriptors, grouped by domain: Time, Frequency Power, Frequency Structure, and Frequency Ratios. Neuro-KE achieves consistently strong linear correlation across most features, particularly for relative band power and cross-frequency ratio biomarkers, demonstrating reliable recovery of interpretable EEG knowledge targets during masked modeling pre-training.

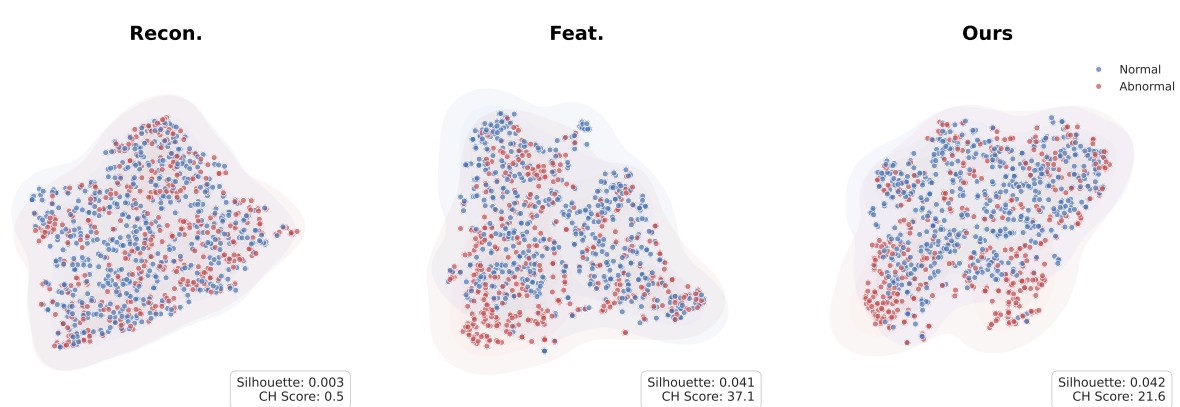

*Figure 12.* **t-SNE visualization of learned EEG representations.** We visualize encoder embeddings pre-trained on the large-scale TUEG corpus and project them into 2D using t-SNE. Points are colored by TUAB abnormality labels (Normal vs. Abnormal), despite the model never being trained with TUAB supervision. Compared to the reconstruction baseline (**Recon.**), both feature-only pretraining (**Feat.**) and Neuro-KE (**Ours**) exhibit improved class-aware clustering structure, indicating that knowledge-guided objectives inject semantically meaningful separation into the learned latent space even under label-free pre-training.

