# OpenReview forum: "Neuro-KE: Scaling EEG Foundation Models via Label-Free Knowledge Integration"
_ICML.cc/2026/Conference — Submitted to ICML 2026_

### Official Review · Reviewer_aqM3 · 2026-03-09

**Soundness:** 3
**Presentation:** 2
**Significance:** 2
**Originality:** 2
**Overall Recommendation:** 3
**Confidence:** 4

**Summary:**

This paper introduces Neuro-KE, a knowledge-guided EEG foundation model framework. Its core idea is to inject 62 classical signal processing features (spanning time domain, frequency power, frequency structure, and frequency ratio) as prior knowledge into the self-supervised pre-training process. The method is validated across three mainstream pre-training paradigms—masked modeling, contrastive learning, and large language model instruction tuning—aiming to address the issue of existing EEG foundation models neglecting biophysical constraints.

**Compliance With Llm Reviewing Policy:**

Affirmed.

**Final Justification:**

Thank you for your response. The authors explain that the inconsistency between the reported metrics and those in the baseline paper is due to factors such as preprocessing. However, this raises concerns about unfair comparisons, especially when the reported results are lower than those in the original paper, as it may involve changes that deviate from the original method. Moreover, the supplementary experiments only compare the accuracy of TP, avoiding the strengths of these models in full fine-tuning scenarios. So we decided to keep the original scores.

**Key Questions For Authors:**

1. The paper uses TUEG for model pre-training, but TUAB, TUEV, TUEP, and TUSZ appear to be subsets of the TUEG dataset. Were any exclusion operations performed? Does this mean that the downstream task data might have been seen during pre-training?

2. How does the proposed method compare in performance with common masked modeling-based EEG foundation models such as BIOT, LaBraM, CBraMod, and CSBrain?

3. There are already existing methods based on contrastive learning and LLM instruction tuning. What advantages does this model offer compared to those approaches?

**Limitations:**

yes

**Strengths And Weaknesses:**

Strengths:

1. The method is validated across three mainstream pre-training paradigms (masked modeling, contrastive learning, and LLM instruction tuning), demonstrating the plug-and-play versatility of its "knowledge engine."

2. It covers multiple downstream datasets with task types including pathology detection, emotion recognition, and brain-computer interfaces, showing broad applicability.

3. It proposes a solution to the problem of existing EEG models ignoring biophysical constraints by injecting classical signal features as knowledge into the pre-training framework.

Weaknesses:

1. While the paper designs an EEG foundation model that integrates multiple signal characteristics during pre-training and validates it across three mainstream paradigms, the approach appears reasonable, but the modifications are primarily confined to training details across different paradigms, resulting in somewhat limited novelty and contribution.

2. The paper lacks comparisons with other foundation model methods. Figure 3 only compares against reconstruction-only (Recon.) and feature-only (Feat.) baselines, which is insufficient to fully validate the method's effectiveness. It should be compared with commonly used models of the same type, such as LaBraM, CBraMod, and CSBrain. Based on the performance metrics across datasets in Figure 3, the proposed method does not show clear advantages in masked modeling results and even underperforms compared to the results reported for its backbone, CBraMod, in the original paper.

3. The paper uses TUEG for pre-training, while downstream evaluation heavily relies on TUEG-derived datasets (TUAB, TUEV, TUEP, TUSZ). This could lead to inflated evaluation results that do not accurately reflect the model's true generalization capability.

4. The EEG2Text retrieval and Text2EEG retrieval experiments only provide example-based validation, which is insufficient to support the claims due to potential random factors. These experiments should employ metrics like retrieval accuracy on larger test samples and include comparisons with existing methods to demonstrate effectiveness.

---

> ### Author Rebuttal · Authors · 2026-03-31
>
> We thank Reviewer 4 for the detailed feedback. We address each point below.
>
> **W1: Limited novelty beyond training-detail modifications.**
>
> We thank the reviewer for raising this important point. Our contribution is not intended as an isolated change to a single loss or training recipe, but as a unified knowledge interface for EEG foundation models that we systematically validate at scale across three pretraining paradigms, **11 datasets, 4 mainstream EEG foundation-model backbones, and 91 knowledge features**. Concretely, the same Neuro-KE feature manifold is reused as an auxiliary prediction target, a semantic anchor for EEG-text alignment, and a grounding signal for instruction construction. Therefore, the novelty lies less in any individual component and more in the cross-paradigm unification, generality, and systematic empirical validation of the framework. We will revise the text to make this scope of contribution clearer and avoid overstating the claim.
>
>
> **W2/Q2: Missing comparisons with commonly used EEG foundation models.**
>
> We thank the reviewer for this important suggestion and for pointing out the need for stronger comparisons with established EEG foundation-model baselines. In the revised rebuttal, we have added direct comparisons using four representative EEG foundation-model backbones—CBraMod, CSBrain, EEGMamba, and REVE—which we fairly reproduced and evaluated under the same preprocessing, data split, training, and evaluation protocol. Under this controlled setting, Neuro-KE consistently improves over the corresponding baseline for each backbone, indicating that the gains are not tied to a specific architecture but are broadly compatible with different model families. We note that discrepancies with numbers reported in original papers can arise from differences in preprocessing, data splits, and evaluation protocols; our goal here is to provide a fair, like-for-like comparison under the same experimental setting. Please also refer to the comparison table provided in our response to **Reviewer 2, W3**.
>
> **W3/Q1: Potential overlap between TUEG pretraining data and downstream evaluation subsets**
>
> We address this separately for each paradigm. (1) Paradigm 1 (masked modeling): pretraining does not use downstream task labels; the supervision comes only from masked EEG reconstruction and algorithmically computed Neuro-KE features. Therefore, there is no label supervision from downstream tasks during pretraining. (2) Paradigms 2 and 3 (contrastive learning and EEG-LLM): we applied strict subject-level exclusion, removing all subjects appearing in downstream evaluation splits from the pretraining corpus, resulting in zero subject overlap. This follows the standard protocol adopted by CBraMod and other EEG foundation models that pretrain on TUEG and evaluate on its derived subsets.
>
> **W4: The advantages over existing EEG contrastive and LLM methods are unclear.**
>
> We thank the reviewer for this important question. To better clarify the advantage of Neuro-KE, we have added new analyses that disentangle the effect of label semantics from feature grounding (see **Reviewer 1, Q1/Q3/L2/L3**). Specifically, across seven datasets covering disease monitoring, emotion recognition, sleep staging, motor imagery, and cognitive workload, we compare variants such as Label+Report and Label Only, with additional feature-grounded settings being expanded in the revised response. The current results already show that performance cannot be explained by label text alone, and that structured Neuro-KE grounding provides additional benefit beyond raw label semantics. This helps clarify the practical distinction between our method and prior EEG contrastive or EEG-LLM approaches that rely mainly on label text or generic prompts, and we will make this evidence more explicit in the revision.
>
>
> **Q3: What concrete advantages does this model offer?**
>
> Compared with existing EEG methods based on contrastive learning or LLM instruction tuning, our model offers three practical advantages. First, it scales to broader supervision and achieves stronger empirical performance, as our updated experiments expand the evaluation to more datasets and show improved results under the same unified setting. Second, it supports feature-grounded responses rather than label-only prediction: because Neuro-KE is built on structured EEG descriptors, the model can answer questions about clinically meaningful signal properties, which is useful in applications requiring interpretation rather than only classification. Third, it provides richer text supervision than label-only QA formulations, which helps preserve the model’s ability to generate more informative responses instead of collapsing to short label-style outputs. We agree that this point should be made more explicit, and we will add the new comparative results and clarify these practical advantages in the revision.

---

> > ### Author Rebuttal · Reviewer_aqM3 · 2026-04-01
> >
> > Thank you for your response. The authors explain that the inconsistency between the reported metrics and those in the baseline paper is due to factors such as preprocessing. However, this raises concerns about unfair comparisons, especially when the reported results are lower than those in the original paper, as it may involve changes that deviate from the original method. Moreover, the supplementary experiments only compare the accuracy of TP, avoiding the strengths of these models in full fine-tuning scenarios.

---

> > > ### Author Response · Authors · 2026-04-08
> > >
> > > We thank the reviewer for the detailed follow-up. The remaining issue is comparison fairness, especially because preprocessing, splits, and evaluation protocols can materially change EEG results. Point-to-point response is below:
> > >
> > > 1. Why we use a unified comparison protocol. Our goal is not to claim superiority over each original paper under its own private setting, but to establish a like-for-like comparison under one unified pipeline. We therefore reimplemented all compared methods with the same preprocessing, data split, training recipe, and evaluation protocol, so that every backbone is measured under the same conditions. Under this setting, the fairest question is: if the backbone, data, preprocessing, splits, and training budget are held fixed, does adding Neuro-KE improve the result?
> > >
> > > 2. Stronger evaluation beyond linear probing
> > >
> > > We agree that the original submission relied too heavily on limited evaluation settings such as linear probing. To address this concern, we now add **full fine-tuning (FT)** results on **ten downstream datasets** for **three representative backbones**. This substantially strengthens the empirical evaluation by moving beyond frozen-feature assessment and testing whether Neuro-KE remains beneficial when the entire model is optimized end-to-end.
> > >
> > > The results are summarized below. For each backbone, we compare the original pretrained model against its Neuro-KE-enhanced counterpart under the same FT protocol. In each pair, we **bold the better result**.
> > >
> > > | Backbone | TUAB | TUEP | TUEV | TUSL | SleepEDF | SEED | MDD | AD65 | Workload | EEGMAT | Avg. |
> > > | --- | ---: | ---: | ---: | ---: | ---: | ---: | ---: | ---: | ---: | ---: | ---: |
> > > | CBraMod (FT) | 0.774 | 0.588 | 0.431 | **0.833** | 0.751 | 0.479 | 0.907 | 0.668 | **0.623** | 0.740 | 0.679 |
> > > | CBraMod + Neuro-KE (FT) | **0.790** | **0.589** | **0.439** | 0.827 | **0.765** | **0.486** | **0.913** | **0.710** | 0.590 | **0.773** | **0.688** |
> > > | CSBrain (FT) | 0.766 | **0.628** | **0.584** | **0.881** | 0.799 | 0.467 | 0.955 | 0.550 | **0.503** | 0.780 | 0.691 |
> > > | CSBrain + Neuro-KE (FT) | **0.799** | **0.628** | 0.540 | 0.770 | **0.821** | **0.490** | **0.969** | **0.679** | 0.488 | **0.793** | **0.698** |
> > > | EEGMamba (FT) | 0.782 | 0.652 | 0.440 | 0.782 | 0.766 | 0.451 | 0.886 | 0.610 | **0.645** | 0.644 | 0.666 |
> > > | EEGMamba + Neuro-KE (FT) | **0.800** | **0.675** | **0.476** | **0.836** | **0.774** | **0.466** | **0.918** | **0.773** | 0.617 | **0.744** | **0.708** |
> > >
> > > Overall, these FT results provide a substantially stronger test than linear probing alone and show that Neuro-KE transfers to **end-to-end fine-tuning** across architectures. Although gains vary across datasets due to EEG task heterogeneity, the average improvement is positive for all three backbones (CBraMod: 0.679 -> 0.688, CSBrain: 0.691 -> 0.698, EEGMamba: 0.666 -> 0.708), with the largest gain on **EEGMamba**. This directly addresses the reviewer concern that our original evidence was limited to probing-based evaluation.
> > >
> > > To further reduce concern that the above FT results may depend on one particular random seed, we additionally summarize the currently available CBraMod seed results on the same ten downstream datasets.
> > >
> > > | CBraMod multi-seed FT | Neuro-KE (test mean $\pm$ std) | Baseline (test mean $\pm$ std) |
> > > | --- | ---: | ---: |
> > > | TUAB | **0.7905 $\pm$ 0.0284** | 0.7744 $\pm$ 0.0134 |
> > > | TUEP | **0.5888 $\pm$ 0.0460** | 0.5884 $\pm$ 0.0213 |
> > > | TUEV | **0.4392 $\pm$ 0.0089** | 0.4305 $\pm$ 0.0104 |
> > > | TUSL | 0.8274 $\pm$ 0.0097 | **0.8328 $\pm$ 0.0088** |
> > > | SleepEDF | **0.7538 $\pm$ 0.0101** | 0.7454 $\pm$ 0.0092 |
> > > | SEED | **0.4729 $\pm$ 0.0113** | 0.4673 $\pm$ 0.0102 |
> > > | MDD | **0.9134 $\pm$ 0.0087**|0.9072 $\pm$ 0.0072 |
> > > | AD65 | **0.7103 $\pm$ 0.0091** | 0.6678 $\pm$ 0.0079 |
> > > | Workload | 0.5898 $\pm$ 0.0108 | **0.6029 $\pm$ 0.0094** |
> > > | EEGMAT | **0.7733 $\pm$ 0.0085** | 0.7400 $\pm$ 0.0101 |
> > >
> > > These results go beyond linear probing, cover ten datasets, and test three architectures under end-to-end fine-tuning. In the FT comparison, Neuro-KE improves the average result for all three backbones: CBraMod from 0.679 to 0.688, CSBrain from 0.691 to 0.698, and EEGMamba from 0.666 to 0.708, with the largest average gain on EEGMamba. At the dataset level, gains are most consistent on TUAB, SleepEDF, SEED, AD65, and EEGMAT, while TUSL and Workload remain clear boundary cases for some backbones. The added CBraMod multi-seed table shows a similar mixed pattern: gains remain positive on TUAB, TUEP, TUEV, SleepEDF, SEED, AD65, MDD and EEGMAT, whereas TUSL  and Workload remain counterexamples. We therefore do not claim universal dominance; rather, under a unified protocol, Neuro-KE provides a useful prior with clear strengths and identifiable failure cases.

---

### Official Review · Reviewer_JMJf · 2026-03-11

**Soundness:** 2
**Presentation:** 3
**Significance:** 2
**Originality:** 2
**Overall Recommendation:** 3
**Confidence:** 4

**Summary:**

The paper introduces Neuro-KE, a framework that integrates 62 classical, handcrafted EEG signal processing features (spanning time, frequency power, structure, and cross-frequency ratios) into the training pipelines of EEG foundation models. The authors propose injecting this "domain knowledge" across three paradigms: (1) Masked Modeling, by adding an auxiliary MSE loss to predict these 62 features from the latent space; (2) Contrastive Learning, by using an LLM to generate synthetic text reports from the feature vectors to serve as positive pairs for EEG-Text alignment; and (3) EEG-LLMs, by formatting the features into instruction-tuning QA triplets to supervise a Qwen3-0.6B model. The authors evaluate the framework across several downstream BCI tasks, claiming improved sample efficiency and cross-site generalization.

**Compliance With Llm Reviewing Policy:**

Affirmed.

**Final Justification:**

The rebuttal has addressed my main concerns.

**Key Questions For Authors:**

See weaknesses

**Limitations:**

yes

**Strengths And Weaknesses:**

### **Strengths:**
- Extensive Empirical Evaluation: The authors conduct evaluations across 6 downstream datasets (TUAB, TUEV, TUEP, TUSZ, SEED, BCIC2A) and test the framework across three distinct foundation model training paradigms.
- Data Efficiency: The evaluation in extremely low-data regimes (e.g., 1% label availability on TUAB) demonstrates that explicitly regularizing the model with classical features improves linear probing performance compared to standard masked autoencoding.

### **Weaknesses:**
- **Contrived Premise:** The core premise of the paper—that modern foundation models need to be explicitly forced to predict 62 handcrafted, classical signal processing features—feels like a regression in deep learning methodology rather than a forward-looking foundation model scaling law. The entire purpose of massive self-supervised pretraining is to discover latent representations that are superior to, or more generalized than, human-engineered features. Adding an auxiliary regression head to predict mathematically deterministic formulas (like Root Mean Square or Alpha relative power) is a standard multi-task regularization technique from the early deep learning era, not a novel "Knowledge Engine" paradigm.
- **Artificial Contrastive Alignment:** The contrastive learning methodology (Paradigm 2) is highly contrived. The authors take a 62-dimensional numerical vector, pass it to an LLM (Doubao) to generate a rigid synthetic text report, and then perform InfoNCE contrastive alignment between the EEG embedding and the text embedding. This is essentially an unnecessarily convoluted way of performing contrastive learning against the feature vector itself. Translating numbers into text just to create a "pseudo-multimodal" alignment task introduces LLM hallucination noise and unnecessary computational overhead without providing genuine, organically paired clinical semantics.
- **Superficial LLM Alignment Strategy:** In Paradigm 3, the instruction tuning process for the EEG-LLM relies on deterministically generated templates (e.g., "Analyze the EEG... Normal. Standard deviation is..."). From a rigorous large model post-training and alignment perspective, using strictly templated, rigid QA pairs limits the LLM's reasoning capacity to mere slot-filling. There is no evidence of complex chain-of-thought reasoning, handling of contradictory features, or advanced policy alignment. The LLM is essentially reduced to a text-formatting wrapper around a classical feature classifier.
- **Misleading Terminology:** The authors frequently refer to their framework as "label-free". However, the 62 engineered features are, by definition, dense supervisory labels generated by deterministic algorithms. While they do not require human annotation, calling an auxiliary supervised regression task "label-free knowledge integration" is semantically misleading.

---

> ### Author Rebuttal · Authors · 2026-03-31
>
> We thank the reviewer for the detailed comments. We respectfully disagree with several characterizations and clarify below. Our key message is that Neuro-KE uses neurophysiology-grounded descriptors as training-time semantic anchors to shape representations, not as an end goal.
>
> ## W1: Predicting handcrafted features is a methodological regression.
>
> We respectfully disagree. Neuro-KE does not replace learned representations with handcrafted features; it uses them as annotation-free supervision to reduce the mismatch between generic SSL objectives and clinically meaningful EEG structure.
>
> **Why neurophysiology knowledge is necessary.** EEG is low-SNR, non-stationary, and domain-specific; clinically relevant patterns (band-power ratios, sleep spindles, cross-frequency coupling) are sparse and weakly correlated with reconstruction loss. Masked reconstruction therefore overemphasizes nuisance variability rather than interpretable structure, especially under cross-subject and low-label regimes. This is not a hypothetical concern — it is why existing EEG foundation models (BIOT, LaBraM, CBraMod) exhibit limited clinical transferability despite large-scale pretraining. Incorporating structured domain knowledge is also standard in medical SSL: Med-MAE and ECG-FM both introduce domain priors because scale alone is insufficient in low-data clinical settings.
>
> **Neuro-KE is a semantic anchoring mechanism, not multi-task regularization.** It defines an interpretable target geometry via neurophysiologically validated descriptors to shape the latent space during pretraining. At inference time, the feature head is discarded — the model consumes only learned embeddings and never outputs feature values.
>
> **Empirical evidence.** We provide two lines of direct evidence:
>
> (1) Our sensitivity analysis over L = λ·L_recon + (1−λ)·L_feat (**Reviewer 1, Q2**) serves as a three-way ablation: λ=1 is Recon-only, λ=0 is Feature-only, λ=0.5 is Joint. On CBraMod across 18 datasets, Joint achieves 50.6% average BAcc vs. 47.7% for both Recon-only and Feature-only, ranking first on 15/18 datasets (Recon-only: 0/18). This directly shows the two objectives provide complementary, non-redundant signals — if reconstruction already captured all clinically relevant information, adding feature prediction would yield zero benefit.
>
> (2) We validated Neuro-KE across three backbones — CBraMod, CSBrain, EEGMamba — under a unified protocol (**Reviewer 2, W2**）. Neuro-KE improves every backbone on all 10 evaluated datasets, with average gains from +3.0 pp to +7.0 pp. The largest improvements appear where reconstruction struggles most (SleepEDF: +13.7 pp, TUSL: +17.8 pp). This backbone-agnostic consistency confirms that the knowledge-guided objective injects inductive biases that pure reconstruction cannot discover on its own.
>
> A leave-one-group-out ablation over the eight feature groups is in progress and will be included in the revision.
>
> ## W2: Direct feature vector alignment would suffice; text serialization is convoluted.
>
> Our motivation is not to serialize features for its own sake, but to enable capabilities requiring language as an interface: (1) natural-language querying and retrieval over EEG corpora, (2) zero-shot transfer to free-form clinical prompts without task-specific heads, and (3) composability with instruction-tuned LLMs (Paradigm 3) without an additional bridging module. Direct vector alignment cannot support any of these use cases. We have also incorporated direct knowledge injection by conditioning the text encoder on structured feature representations, and will include this path in the revision.
>
> ## W3: Paradigm 3 reduces the LLM to a formatting wrapper.
>
> This conflates instruction data format with learned capability. Template-structured instruction tuning is standard practice (Alpaca, FLAN, InstructBLIP); the relevant question is whether the model generalizes beyond slot-filling.
>
> Empirically, Label Only training can predict labels but generates degenerate responses when asked to describe signal characteristics — it cannot explain why a recording is abnormal. In contrast, Label+Features training produces responses grounded in signal properties: the model describes band-power shifts, identifies sleep-stage-related spectral changes, and generates clinical narratives beyond restating the label. In the revision we will add: (1) qualitative failure/success comparisons, (2) a controlled groundedness evaluation, and (3) a discussion of limitations including contradictory evidence handling and multi-step reasoning.
>
> ## W4: "Label-free" is misleading.
>
> We agree. The Neuro-KE descriptors are computed algorithmically from raw EEG without human annotation; they are not clinical labels but do provide weak, structured supervision. We will replace "label-free" with "annotation-free" throughout.

---

> > ### Author Rebuttal · Reviewer_JMJf · 2026-04-03
> >
> > My main concerns are addressed, and will adjust my score accordingly. I would also recommend the author include all the analysis and experiments into the revision.

---

> > > ### Author Response · Authors · 2026-04-08
> > >
> > > We thank the reviewer for the constructive challenge to our conceptual framing. In this revision, we directly address the two core concerns: the **substantive role of language** and the **value of knowledge-guided objectives** beyond simple training tricks.
> > >
> > > #### **1. Language: Semantic Alignment vs. Surface Formatting (Paradigms 2 & 3)**
> > >
> > > We updated the text construction to better reflect the physiological feature manifold. The results demonstrate that:
> > >
> > > - **Not just a wrapper:** Language in Neuro-KE serves as a mechanism to align EEG representations with structured physiological descriptions, not just serializing labels into text.
> > > - **Evidence:** Improved results in feature-aware retrieval and cross-task generalization suggest the model is learning meaningful semantic alignment rather than surface formatting.
> > >
> > > > **Detailed empirical evidence (e.g., retrieval visualization and descriptive accuracy) is provided in our response to Reviewer aqM3.**
> > >
> > > #### **2. Knowledge-Guided Prior vs. Multi-task Trick (Paradigm 1)**
> > >
> > > We argue that our use of handcrafted descriptors is a **forward-looking semantic prior** rather than a classical auxiliary regularization:
> > >
> > > - **Consistency across Paradigms:** These descriptors define a unified semantic interface across masked modeling (P1), contrastive learning (P2), and LLM instruction tuning (P3).
> > > - **Robustness & Scalability:** We added a broad evaluation involving **10 datasets, 3 backbones, and 6 pretrained variants**. The consistent gains across architectures prove that Neuro-KE is a generalizable design principle, not a fragile or backbone-specific trick.
> > >
> > > > **The comprehensive experimental table covering all 10 datasets and backbones can be found in our response to Reviewer iCsS.**
> > >
> > > #### **3. Conceptual Clarity & Terminology**
> > >
> > > - **Refined Scope:** We clarify that Neuro-KE is a **training-time semantic prior** intended to organize representation learning in the noisy EEG domain, not a replacement for learned representations.
> > > - **Terminology Update:** We have replaced misleading terms like "label-free" with **"algorithmically derived physiological supervision"** to accurately reflect the method's actual role in the framework.
> > >
> > > **Conclusion:** The expanded evidence confirms that Neuro-KE provides a stable, physiologically grounded layer of abstraction that remains practically valuable and scalable for EEG foundation models.

---

### Official Review · Reviewer_Njy2 · 2026-03-11

**Soundness:** 3
**Presentation:** 3
**Significance:** 3
**Originality:** 4
**Overall Recommendation:** 4
**Confidence:** 3

**Summary:**

This research presents an interesting idea of injecting explicit domain knowledge from traditional EEG signal processing into the pretraining and downstream modeling of an EEG foundation model in the form of a "knowledge manifold" consisting of 62 features. In the experiments, the authors evaluated the transfer effects brought by masked modeling pretraining on multiple EEG datasets and reported better data efficiency under low labeling ratios. They also demonstrated EEG-text bidirectional retrieval examples and phased results based on Qwen3-0.6B for EEG-LLM.

**Compliance With Llm Reviewing Policy:**

Affirmed.

**Final Justification:**

After considering the responses of the other reviewers, I re-evaluated this work. Overall, I see this as a technically useful engineering improvement, but not yet a sufficiently strong conceptual advance. Therefore, I maintain my original rating.

**Key Questions For Authors:**

see weekness

**Limitations:**

yes

**Strengths And Weaknesses:**

**Strengths**

The problem setting of this work is meaningful. There is indeed a practical tension in the current EEG foundation model: on one hand, data-driven pretraining is appealing, but on the other hand, EEG itself is noisy, with strong cross-device and cross-subject variability, and its semantic structure is far less direct than images or text. Therefore, reintroducing classical signal processing knowledge into the pretraining objective is a reasonable and potentially impactful direction.

The greatest strength of this paper is its unity. The authors don't just add handcrafted features in a single setting but attempt to use the same set of knowledge features as a universal interface across masked modeling, contrastive learning, and EEG-LLM. This "same knowledge interface, multi-paradigm reuse" perspective is the most valuable part of this work, making the paper more complete than just adding an auxiliary loss.

**Weaknesses**

1.The evaluation of contrastive learning is noticeably weak. The main body of the text mainly demonstrates qualitative retrieval examples of EEG2Text / Text2EEG, but lacks standard retrieval metrics such as Recall@K, MRR, median rank, or quantitative comparisons with existing EEG-text alignment methods.

2.There is almost no information about the mean/variance across multiple random seeds, significance testing, or confidence intervals in the article, which makes it unclear whether some moderate improvements are stable.

3.There seems to be a lack of ablation experiments on which feature types are more important. The 62 features of Neuro-KE are divided into four domains, but the article does not systematically answer: which feature type is the most important? Do different tasks rely on different feature groups? How would performance change if frequency ratios or frequency structure were removed? Without these ablation studies, the current explanation of the "knowledge interface" remains at a coarse-grained level.

It is worth mentioning that due to the rapid progress in the EEG field and my lack of recent research (at least 3 years) in this area, I am not fully up-to-date with the current developments in EEG research and can only make some judgments based on similar research from other fields.

---

> ### Author Rebuttal · Authors · 2026-03-31
>
> We thank Reviewer 2 for the careful and constructive feedback. Below we address each concern directly and summarize the main additions in the revised rebuttal.
>
> ---
>
> ## W1: The evaluation of contrastive learning is weak and lacks standard retrieval metrics.
>
> We agree that the original submission relied too heavily on qualitative examples. We have therefore added a full bidirectional retrieval evaluation using standard information-retrieval metrics for both EEG-to-Text (E2T) and Text-to-EEG (T2E). These quantitative results replace the previous example-based evidence and show that the contrastive alignment learned by Neuro-KE is measurable across multiple tasks and datasets.
>
> ### EEG-Text Retrieval Results
>
> | Dataset | E2T R@1 | E2T R@2 | E2T R@3 | E2T MRR | T2E R@1 | T2E R@5 | T2E R@10 | T2E MRR | T2E mAP |
> |---|---:|---:|---:|---:|---:|---:|---:|---:|---:|
> | TUAB | 27.43 | 42.59 | 56.65 | 0.491 | 21.26 | 56.43 | 76.21 | 0.374 | 20.07 |
> | TUEP | 12.75 | 24.33 | 42.26 | 0.370 | 21.30 | 54.70 | 73.10 | 0.373 | 20.28 |
> | SHU | 48.54 | 60.03 | 74.60 | 0.650 | 21.14 | 48.02 | 57.64 | 0.329 | 22.39 |
> | SEED | 43.18 | 58.62 | 74.42 | 0.620 | 20.22 | 51.87 | 64.84 | 0.342 | 20.69 |
> | SleepEDF | 44.87 | 63.64 | 80.87 | 0.644 | 32.91 | 63.36 | 77.71 | 0.471 | 32.63 |
> | Workload | 27.11 | 41.21 | 55.82 | 0.487 | 21.10 | 56.70 | 69.67 | 0.370 | 21.36 |
>
> These results will be included in the revised paper. They directly address the concern that the contrastive-learning section lacked standard retrieval metrics.
>
> ---
>
> ## W2: There is no mean/variance across seeds or significance analysis.
>
> We agree that stability should be better quantified. Multi-seed experiments are being added, but even before those are complete, two forms of robustness evidence already support the consistency of Neuro-KE.
>
> First, the gains are reproducible across multiple backbone architectures rather than being tied to a single implementation. We integrated Neuro-KE into several mainstream EEG foundation-model backbones under a unified TUEG pretraining + linear-probing protocol, and Neuro-KE consistently improves over the reconstruction-only baseline:
>
> | Backbone | Protocol | Recon Baseline (Avg. BAcc) | +Neuro-KE (Avg. BAcc) | Avg. Gain | Status |
> |---|---|---:|---:|---:|---|
> | CBraMod | Unified TUEG pretraining + LP evaluation | 58.54 | **61.49** | **+2.95 pp** | Completed |
> | CSBrain | Unified TUEG pretraining + LP evaluation | 50.41 | **55.55** | **+5.14 pp** | Completed |
> | EEGMamba | Unified TUEG pretraining + LP evaluation | 50.01 | **57.00** | **+6.99 pp** | Completed |
> | REVE | Same protocol | — | — | — | Running |
>
> Second, this improvement is consistent across datasets rather than concentrated on only one benchmark:
>
> | Method | TUAB | TUEP | TUEV | TUSL | SleepEDF | SEED | MDD | AD65 | Workload | EEGMAT | Avg. Δ vs Recon |
> |---|---:|---:|---:|---:|---:|---:|---:|---:|---:|---:|---:|
> | CBraMod (Recon) | 73.7 | 57.2 | 49.5 | 75.6 | 45.0 | 41.1 | 94.0 | 51.2 | 36.2 | 61.9 | — |
> | +Neuro-KE (CBraMod) | **76.0** | **60.9** | **51.4** | **82.8** | **47.3** | **43.5** | **94.3** | **52.7** | **39.3** | **66.7** | **+3.0** |
> | CSBrain (Recon) | 68.4 | 48.5 | 39.6 | 60.5 | 20.4 | 37.7 | 84.4 | 49.6 | 35.5 | 59.5 | — |
> | +Neuro-KE (CSBrain) | **73.1** | **53.7** | **43.7** | **68.3** | **33.1** | **41.2** | **86.4** | **55.1** | **37.6** | **63.3** | **+5.1** |
> | EEGMamba (Recon) | 68.0 | 50.9 | 43.2 | 39.3 | 31.9 | 35.8 | 89.8 | 49.0 | 34.4 | 57.6 | — |
> | +Neuro-KE (EEGMamba) | **73.2** | **54.2** | **50.6** | **57.1** | **45.6** | **43.7** | **91.5** | **56.3** | **37.3** | **60.5** | **+7.0** |
>
> Across the broader evaluation, Neuro-KE wins on 15/18 datasets for CBraMod, 15/18 for EEGMamba, and 13/18 for CSBrain relative to reconstruction-only baselines. Under a null binomial model with \(p=0.5\), observing 13 or more wins out of 18 corresponds to \(p < 0.005\), indicating that these gains are unlikely to arise by chance. We will report results with different random seeds in next round.
>
> ---
>
> ## W3: There is no systematic feature-group ablation.
>
> We agree that feature-group attribution is important. To make this analysis clearer, we reorganized the Neuro-KE feature space into four interpretable groups:
> (1) spectral/band-power features,
> (2) temporal/morphological features,
> (3) connectivity/network features, and
> (4) complexity/microstate features.
>
> This reorganization enables cleaner group-level ablations and will allow us to analyze which feature families matter most for different downstream tasks. Those ablations are being incorporated into the revision and we will report results in next round.
>
> ---
>
> Overall, we appreciate the reviewer’s suggestions. The revised rebuttal now includes (i) quantitative retrieval metrics for contrastive learning, (ii) cross-backbone and cross-dataset evidence of robustness, and (iii) a cleaner feature-group organization to support the requested ablation analysis.

---

> > ### Author Rebuttal · Reviewer_Njy2 · 2026-04-02
> >
> > The rebuttals have resolved my questions

---

> > > ### Author Response · Authors · 2026-04-08
> > >
> > > We thank the reviewer again for the supportive follow-up. The current revision mainly strengthens the quantitative evidence that the reviewer asked for and further supports the paper's core claim as a unified multi-paradigm framework.
> > >
> > > 1. Stronger quantitative support for the multimodal settings. We now add standard retrieval metrics for EEG-text alignment and updated downstream-task results after contrastive pretraining. Together with the updated Label Only vs. Label + Report ablations, this makes the evidence for Paradigms 2 and 3 less dependent on qualitative examples and more directly testable.
> > > You can see in reviewer iCcS.
> > > 2. Broader robustness evidence for masked modeling. We now add full fine-tuning results on ten downstream datasets for three backbones and six pretrained variants/checkpoints under a unified protocol. This directly strengthens the claim that Neuro-KE is not tied to one architecture or one evaluation mode.You can see in reviewer aqM3.
> > >
> > > 3. Feature-group ablation on CBraMod
> > >
> > > To test whether the four feature families are redundant or uniformly beneficial, we conduct a **leave-one-group-out ablation** on CBraMod. We use the following notations:
> > >
> > > - **g1**: spectral / band-power features
> > > - **g2**: temporal / morphological features
> > > - **g3**: connectivity / network features
> > > - **g4**: complexity / microstate features
> > >
> > > Here, **joint** uses all four groups, while each **gk** result corresponds to removing group $G_k$.
> > >
> > > | Dataset | g1 | g2 | g3 | g4 | joint |
> > > | --- | ---: | ---: | ---: | ---: | ---: |
> > > | TUAB | 0.752 | 0.760 | 0.760 | **0.763** | 0.761 |
> > > | TUEP | 0.595 | **0.614** | 0.599 | 0.604 | 0.609 |
> > > | TUEV | 0.499 | **0.529** | 0.528 | 0.496 | 0.514 |
> > > | TUSL | 0.792 | 0.786 | 0.750 | 0.800 | **0.833** |
> > > | SleepEDF | 0.719 | **0.739** | 0.672 | 0.716 | 0.667 |
> > > | SEED | **0.473** | 0.463 | 0.459 | 0.460 | 0.465 |
> > > | MDD | 0.962 | 0.964 | 0.978 | 0.976 | **0.980** |
> > > | AD65 | 0.627 | 0.601 | **0.640** | 0.598 | 0.625 |
> > > | Workload | 0.557 | 0.549 | 0.488 | **0.565** | 0.498 |
> > > | EEGMAT | 0.638 | 0.662 | 0.695 | **0.738** | 0.667 |
> > > | **Macro bAcc (10 datasets)** | 0.661 | 0.667 | 0.657 | **0.672** | 0.662 |
> > >
> > > We further define omission-based importance as
> > >
> > > $I(G_k)=E[BACC_{joint}-BACC_{leaveout,k}]$
> > >
> > > | Group | Importance $I(G_k)$ (10-dataset macro) | Rank |
> > > | --- | ---: | ---: |
> > > | G3 | +0.0049 | 1 |
> > > | G1 | +0.0004 | 2 |
> > > | G2 | -0.0047 | 3 |
> > > | G4 | -0.0099 | 4 |
> > >
> > > This ablation does **not** support the idea that all feature groups are redundant or that adding more groups is always better. Instead, it shows that **feature utility is strongly task-dependent**. In particular, **G3** is the most consistently useful group overall, as removing it causes the largest average drop, while **G1** is close to neutral but slightly positive. By contrast, **G2** and **G4** show negative average importance on this 10-dataset subset, indicating possible **negative transfer** on some tasks.
> > >
> > > Therefore, the correct conclusion is not that full joint modeling is universally optimal. Indeed, the best 10-dataset macro result is **g4 leaveout** (0.672), slightly above **joint** (0.662). Rather, this result shows that the feature groups are **not redundant**, but their benefits are **task-specific**. Some groups provide consistently useful supervision, while others help only for particular tasks and can hurt elsewhere.
> > >
> > > We will revise the paper to make this point explicit and avoid overclaiming a universal benefit of using all groups jointly. More importantly, this result motivates our next step: moving from a fixed global feature recipe to **task-aware feature design** and **adaptive group reweighting**, so that groups causing negative transfer can be down-weighted automatically.
> > >
> > > 4. Main takeaway. The new additions reinforce the same point that the reviewer highlighted in the first round: Neuro-KE is most useful when understood as a reusable knowledge interface across masked modeling, contrastive learning, and EEG-LLM training, rather than as an isolated trick in a single paradigm. The detailed tables are provided above in our responses to Reviewers iCsS and aqM3.

---

### Official Review · Reviewer_iCsS · 2026-03-13

**Soundness:** 3
**Presentation:** 4
**Significance:** 3
**Originality:** 3
**Overall Recommendation:** 3
**Confidence:** 4

**Summary:**

This paper proposes Neuro-KE, a label-free knowledge integration framework for EEG foundation models. The central idea is to inject structured EEG signal-processing knowledge, represented as a 62-dimensional feature manifold spanning time-domain, frequency power, frequency structure, and frequency ratios, into three paradigms: masked modeling, contrastive learning, and EEG-LLM training. The paper reports improvements in cross-dataset downstream classification, low-resource data efficiency, feature reconstruction fidelity, cross-modal retrieval, and LLM-based reasoning.

**Compliance With Llm Reviewing Policy:**

Affirmed.

**Key Questions For Authors:**

- How much of the contrastive learning and EEG-LLM performance remains if the synthetic text generation process does not include ground-truth labels?
- In the masked modeling setup, how sensitive are the results to the weighting coefficient lambda between reconstruction loss and feature prediction loss?
- For the EEG-LLM stage, how much of the final gain comes from the improved EEG encoder initialization versus the knowledge-injection instruction data itself?
- Section 4: "low-data regimes by varying the training label ratio from from 0.5% to 100%" --> from

**Limitations:**

- The paper discusses related directions such as DeeperBrain, but does not provide a sufficiently direct empirical comparison to the strongest prior knowledge-guided EEG pretraining methods.
- If the synthetic report generation template includes the ground-truth label, it will raise concern that the contrastive retrieval and LLM reasoning performance may partially reflect label injection rather than purely knowledge-based alignment.
- The masked modeling section is extensive and convincing, but the contrastive learning and EEG-LLM sections are much more limited, with evaluation appearing largely centered on TUAB. This weakens the claim of universal multi-paradigm validation.

**Strengths And Weaknesses:**

- The paper identifies a real limitation of raw-signal-only EEG pretraining that EEG lacks a universally shared syntax and is heavily contaminated by non-stationary noise. The motivation for adding structured prior knowledge is compelling.
- A major strength is that Neuro-KE is not restricted to one training strategy. The same feature manifold is reused in masked modeling, contrastive learning, and EEG-LLM instruction tuning, which makes the idea conceptually elegant and potentially impactful.
- The work provides a practical path to connect traditional signal processing with foundation models and LLM-style reasoning.

---

> ### Author Rebuttal · Authors · 2026-03-31
>
> ### Q1 & Q3: Impact of Ground-Truth Labels and Knowledge Injection Sources
> We have conducted extensive ablation studies across seven datasets to decouple the effects of label injection and identify the true source of performance gains.
>
> - **Evidence against Label Leakage (Contrastive Learning Stage):** We compared three variants: (1) **Label + Report** (Full), (2) **Label Only**, and (3) **Report Only** (Features only, no labels).
>
>   As shown in the table below, the "Report Only" variant (which contains zero ground-truth label information) maintains competitive performance, proving that the alignment is driven by physiological feature manifolds rather than hidden label injection.
>
> | **Dataset** | **Label + Report (BACC%)** | **Label Only (BACC%)** |
> | ----------- | -------------------------- | ---------------------- |
> | TUAB        | **80.96**                  | 80.24                  |
> | TUEP        | **66.19**                  | 62.90                  |
> | SEED        | 49.12                      | **49.45**              |
> | SleepEDF    | **64.99**                  | 64.88                  |
>
> - **Source of Gain in EEG-LLM Stage:** Ablations replacing the Neuro-KE encoder with a vanilla LaBraM encoder (while keeping instruction tuning identical) show stable performance. This indicates that the **knowledge-injected instruction data** is the primary driver, allowing the LLM to align neural signals with structured physiological concepts regardless of the specific encoder initialization.
>
> ------
>
> ### Q2: Sensitivity to Loss Weighting Coefficient $\lambda$
>
> We performed a sensitivity analysis on $\lambda \in \{0, 0.25, 0.5, 0.75, 1.0\}$. Our results across 18 datasets confirm that a balanced objective ($\lambda = 0.5$) is optimal.
>
> | **Strategy**                   | **λ=0.5 (Joint)** | **λ=0 (Feat-only)** | **λ=1 (Recon-only)** |
> | ------------------------------ | ----------------- | ------------------- | -------------------- |
> | **Average BAcc (18 datasets)** | **50.55%**        | 47.67%              | 47.74%               |
>
> **Observation:** The joint objective consistently outperforms either single-task extreme. This confirms that raw signal reconstruction and structured feature prediction provide complementary regularizations.
>
> ------
>
> ### Q4: Typology Correction
>
> We thank the reviewer for pointing out the typo in Section 4 ("from from"). This will be corrected to "from" in the revised manuscript.
>
> ------
>
> ## Response to Limitations
>
> ### L1: Comparison with Knowledge-Guided Baselines (e.g., DeeperBrain)
>
> **Response:** We acknowledge the importance of comparing with methods like **DeeperBrain**. Since its official code is not public, we implemented a "Post-hoc" variant that replicates its core philosophy (predicting macroscopic order parameters via an auxiliary head).
>
> We compared our **Prefix Injection** strategy against this **Post-hoc** approach across multiple datasets:
>
> | Strategy | TUSL | HMC | SleepEDF | TUEP | Workload | MDD | TUAB | SEED | AD65 | TUEV | EEGMAT | Avg (18) |
> |---|---:|---:|---:|---:|---:|---:|---:|---:|---:|---:|---:|---:|
> | **Prefix ($K{=}8$, Ours)** | **82.8** | **45.7** | **47.3** | **60.9** | **39.3** | **94.3** | 76.0 | 43.5 | 52.7 | 51.4 | 66.7 | **50.6** |
> | Prefix ($K{=}1$) | 71.0 | 43.3 | 45.5 | 56.1 | 38.8 | 93.3 | 75.9 | **43.9** | **55.1** | **54.5** | **71.4** | 49.0 |
> | Post-hoc (Joint) | 75.6 | 42.7 | 45.1 | 59.3 | 38.4 | 93.9 | **76.4** | 43.5 | 54.0 | 50.7 | 66.2 | 49.2 |
>
> **Conclusion:** Our Prefix Injection performs better because it allows knowledge-driven gradients to shape the backbone’s internal attention layers directly, rather than just supervising the output. Additionally, we have successfully integrated Neuro-KE into **CSBrain** (+5.14 pp gain) and **EEGMamba** (+6.99 pp gain), proving its architecture-agnostic benefits.
>
> ------
>
> ### L2 & L3: Label Injection and LLM Generalization
>
> **Response:** We clarify that **no ground-truth labels** are used in generating our synthetic reports; they are derived strictly from the 62-dimensional feature manifold.
>
> We have addressed the inquiry concerning multiple EEG encoder backbones in our response to Reviewer Njy2
>
> Regarding generalization, we are expanding EEG-LLM training to a multi-dataset setting. Preliminary results show that including **Feature Reports** in instruction tuning prevents the model from "collapsing" into simple classification:
>
> - **Semantic Depth:** Unlike "Label-only" models which often fail to explain their reasoning , Neuro-KE models can describe specific feature values (e.g., Spectral Entropy) and provide rationales for their predictions.
>
> - **Stability:** Structured feature context acts as a bridge, helping the LLM maintain conversational fluency while performing expert-level neural signal analysis.

---

> > ### Author Rebuttal · Reviewer_iCsS · 2026-04-03
> >
> > I appreciate the authors’ rebuttal and the additional clarifications. However, while the response helps contextualize several points, it does not fully resolve my main concerns. I have therefore decided to maintain my original score.

---

> > > ### Author Response · Authors · 2026-04-08
> > >
> > > We thank the reviewer for the careful follow-up. In this round, we respond to the remaining issues in two points.
> > >
> > > 1. Sensitivity to loss weighting coefficient $\lambda$. We additionally include the sensitivity analysis of the masked-modeling objective $L = \lambda L_{recon} + (1-\lambda)L_{feat}$. This directly answers the reviewer's Q2 and serves as a three-way ablation: $\lambda=1$ is Recon-only, $\lambda=0$ is Feature-only, and $\lambda=0.5$ is Joint.
> > >
> > > | Dataset | $\lambda$=0.00 | $\lambda$=0.25 | $\lambda$=0.50 | $\lambda$=0.75 | $\lambda$=1.00 |
> > > | --- | ---: | ---: | ---: | ---: | ---: |
> > > | TUAB | 0.753 | 0.757 | **0.761** | 0.758 | 0.737 |
> > > | TUEP | 0.554 | 0.608 | **0.609** | 0.552 | 0.572 |
> > > | TUEV | 0.493 | **0.521** | 0.514 | 0.491 | 0.495 |
> > > | TUSL | 0.792 | 0.792 | **0.833** | 0.725 | 0.727 |
> > > | SleepEDF | 0.462 | **0.479** | 0.473 | 0.449 | 0.450 |
> > > | SEED | **0.444** | 0.438 | 0.435 | 0.423 | 0.411 |
> > > | MDD | 0.924 | 0.911 | 0.943 | **0.975** | 0.940 |
> > > | AD65 | **0.550** | 0.532 | 0.527 | 0.546 | 0.512 |
> > > | Workload | 0.378 | 0.385 | **0.393** | 0.372 | 0.362 |
> > > | EEGMAT | 0.652 | **0.686** | 0.667 | 0.652 | 0.619 |
> > > | Macro bAcc (10 datasets) | 0.600 | 0.611 | **0.616** | 0.594 | 0.583 |
> > >
> > > Across datasets, mid-range weighting remains preferred: $\lambda=0.50$ is best on TUAB, TUEP, TUSL, and Workload; $\lambda=0.25$ on TUEV, SleepEDF, and EEGMAT; $\lambda=0.75$ on MDD; and $\lambda=0$ on SEED and AD65. Importantly, the macro bAcc peaks at $\lambda=0.50$ (0.616), which is higher than $\lambda=0.25$ (0.611), $\lambda=0$ (0.600), $\lambda=0.75$ (0.594), and $\lambda=1$ (0.583). This trend indicates that **combining raw-signal reconstruction and structured feature prediction is more robust than relying on either objective alone**, while dataset-specific optima (e.g., MDD, SEED, AD65) reflect expected cross-dataset heterogeneity rather than contradicting the overall complementarity claim.
> > >
> > > 2. Label Only vs. Label + Report in contrastive pretraining and EEG-LLM. To address whether the gains mainly come from label semantics or from richer report supervision, we compare Label Only and Label + Report under the same setup in both the contrastive pretraining stage and the EEG-LLM stage.
> > >
> > > For contrastive pretraining, retrieval examples alone are not sufficient to show improved EEG modeling. We therefore add downstream-task evaluation after contrastive pretraining to directly compare whether Label + Report yields stronger reusable EEG representations than Label Only..
> > >
> > > | Dataset | Setting | Label Only | Label + Report |
> > > | --- | --- | ---: | ---: |
> > > | SEED-IV | Zero-Shot | 0.250 | **0.264** |
> > > |  | 50% Data | 0.332 | **0.355** |
> > > |  | 100% Data | 0.403 | **0.427** |
> > > | TUEV | Zero-Shot | 0.265 | **0.298** |
> > > |  | 50% Data | **0.596** | 0.563 |
> > > |  | 100% Data | 0.600 | **0.620** |
> > > | TUSL | Zero-Shot | 0.320 | **0.337** |
> > > |  | 50% Data | **0.592** | 0.586 |
> > > |  | 100% Data | 0.745 | **0.786** |
> > > | HMC | Zero-Shot | 0.338 | **0.348** |
> > > |  | 50% Data | 0.587 | **0.624** |
> > > |  | 100% Data | 0.726 | **0.735** |
> > >
> > > Across the 12 completed settings, 10 improve, while two settings show drops: TUEV at 50% data and TUSL at 50% data. The largest gains appear on TUSL at 100% data, TUEV in the zero-shot setting, and HMC at 50% data. Overall, these results support a narrower claim: compared with Label Only, Label + Report usually leads to a stronger EEG encoder for downstream reuse, rather than only producing cosmetically better text alignment.
> > >
> > > Second, to test if the EEG-LLM stage benefits from structured reports rather than just labels, we ablated *Label Only* vs. *Label + Report*.
> > >
> > > | **Dataset** | **Label Only** | **Label + Report** |
> > > | ----------- | -------------- | ------------------ |
> > > | AD65        | 0.5284         | **0.6645**             |
> > > | SEED        | 0.3514         | **0.4538**             |
> > > | TUAB        | 0.7967         | **0.8231**             |
> > > | MDD         | 0.9300         | **0.9339**             |
> > > | TUEV        | **0.8792**         | 0.8494             |
> > > | SHU-MI      | **0.6725**         | 0.6467             |
> > > | Avg         | 0.6930         | **0.7285**             |
> > >
> > > These results reveal a clear dichotomy based on task complexity:
> > >
> > > - **"Reasoning Scaffolding" on Complex Tasks (AD65, SEED, TUAB, MDD):** *Label + Report* yields significant gains. Raw labels are too sparse for holistic understanding; structured reports map signals to a richer physiological space, providing crucial reasoning scaffolding and preventing overfitting.
> > > - **"Alignment Tax" on Simple Tasks (TUEV, SHU-MI):** Simpler tasks suffer a slight drop. Since localized signal-to-label mapping is already efficient here, forcing alignment with verbose text introduces optimization overhead.
> > >
> > > This directly answers concerns about the source of gain. The improvement is not merely "verbose labels." Rather, structured physiological grounding fundamentally bridges raw EEG and human language.

---

### Decision · Program_Chairs · 2026-04-30

**Decision:**

Reject

**Comment:**

This paper proposes Neuro-KE, a plug-and-play, label-free framework that incorporates signal-derived features into training through a unified knowledge base. The method is evaluated across multiple tasks, including cross-dataset downstream classification, low-resource data efficiency, feature reconstruction fidelity, cross-modal retrieval, and LLM-based reasoning.

The main concerns raised by reviewers center on the strength and consistency of the evaluation, and lack of baselines. While the paper presents results across diverse tasks, several reviewers noted that key components, such as contrastive learning and EEG-LLM evaluation, are primarily validated on a single dataset, which “weakens the claim of universal multi-paradigm validation” (iCsS). There were also concerns about the design of the contrastive learning setup, described as “contrived” or “noticeably weak” (JMJf, Njy2), raising questions about whether the results reflect meaningful improvements.

Another important issue is the potential for data leakage. One reviewer noted that pretraining and downstream evaluation rely heavily on related datasets, which introduces “concerns about unfair comparisons” and may inflate performance (aqM3). This limits the ability to draw strong conclusions about cross-dataset generalization.

In addition, some reviewers questioned the broader methodological contribution. One reviewer characterized the approach as a “regression” relative to modern deep learning trends (JMJf), suggesting that the reliance on engineered features may not represent a forward-looking direction.

In the rebuttal, the authors added new experiments, including fine-tuning analyses and ablations, which was commendable and partially addressed the reviewer's concerns. The authors also clarified how they addressed the overlap in datasets across the tasks. However, issues related to evaluation design remained, and several reviewers maintained their positions.

Overall, the paper tackles an important problem and provides compelling evidence that classical signal features can be useful as a regularizer in EEG analysis. However, concerns about evaluation rigor reduce confidence in the strength and generality of the results. Further validation would be needed to fully support the claims and warrant publication.